# Managing urban development could halve nitrogen pollution in China

Ouping Deng[1,2,11], Sitong Wang[1,3,11], Jiangyou Ran[2], Shuai Huang[2], Xiuming Zhang [1], Jiakun Duan[1], Lin Zhang [4], Yongqiu Xia[5], Stefan Reis[6], Jiayu Xu [4], Jianming Xu [1,7], Wim de Vries [8], Mark A. Sutton[9] & Baojing Gu [1,3,10] ✉

Halving nitrogen pollution is crucial for achieving Sustainable Development Goals (SDGs). However, how to reduce nitrogen pollution from multiple sources remains challenging. Here we show that reactive nitrogen (Nr) pollution could be roughly halved by managed urban development in China by 2050, with $NH_3$, $NO_x$ and $N_2O$ atmospheric emissions declining by 44%, 30% and 33%, respectively, and Nr to water bodies by 53%. While rural-urban migration increases point-source nitrogen emissions in metropolitan areas, it promotes large-scale farming, reducing rural sewage and agricultural non-point-source pollution, potentially improving national air and water quality. An investment of approximately US$ 61 billion in waste treatment, land consolidation, and livestock relocation yields an overall benefit of US$ 245 billion. This underscores the feasibility and cost-effectiveness of halving Nr pollution through urbanization, contributing significantly to SDG1 (No poverty), SDG2 (Zero hunger), SDG6 (Clean water), SDG12 (Responsible consumption and production), SDG14 (Climate Action), and so on.

The invention of the Haber-Bosch process for nitrogen fixation (HBNF) has doubled global grain production that has supported rapid growth of the global population[1]. However, reactive nitrogen (Nr) losses during food and energy production and consumption have led to multiple unintended consequences, including air and water pollution, soil acidification, biodiversity loss and global warming[2–4]. The United Nations Environment Programme (UNEP) has formulated the objective to halve nitrogen waste globally to achieve global sustainable development goals (SDGs)[5], which has now been embraced by the UN Convention on Biological Diversity[6]. Nitrogen use and losses are

directly linked to about half of SDGs such as *No poverty* (SDG1), *Zero hunger* (SDG2), *Clean water* (SDG6) and *Responsible consumption and production* (SDG12), *Climate Action* (SDG14), and *Life and Land* (SDG15)[7,8]. Therefore, halving nitrogen waste is key to achieving SDGs in coming decades.

China is a hotspot of global Nr pollution, using about 30% of global nitrogen fertilizers and feeding 22% of global livestock[9]. Wastage of Nr from agriculture, industrial production and human activities has potentially contributed to China's air and water quality challenges, leading to millions of premature deaths annually[10–12].

[1]College of Environmental and Resource Sciences, Zhejiang University, Hangzhou 310058, China. [2]College of Resources, Sichuan Agricultural University, Chengdu 611130, China. [3]Policy Simulation Laboratory, Zhejiang University, Hangzhou 310058, China. [4]Department of Atmospheric and Oceanic Sciences, School of Physics, Peking University, Beijing 100871, China. [5]Key Laboratory of Soil and Sustainable Agriculture, Changshu National Agr-Ecosystem Observation and Research Station, Institute of Soil Science, Chinese Academy of Sciences, Nanjing 210008, China. [6]Unit for Environment and Sustainability at the German Aerospace Centre's Project Funding Agency, DLR Projekttraeger, Bonn 53227, Germany. [7]Zhejiang Provincial Key Laboratory of Agricultural Resources and Environment, Zhejiang University, Hangzhou 310058, China. [8]Environmental Systems Analysis Group, Wageningen University & Research, Wageningen 91016700HB, The Netherlands. [9]UK Centre for Ecology & Hydrology, Bush Estate, Penicuik, Midlothian EH260QB, UK. [10]Ministry of Education Key Laboratory of Environment Remediation and Ecological Health, Zhejiang University, Hangzhou 310058, China. [11]These authors contributed equally: Ouping Deng, Sitong Wang. ✉e-mail: bjgu@zju.edu.cn

However, it is difficult to reduce nitrogen waste given the multitude of Nr sources (e.g. cropland, livestock, sewage) and Nr forms (e.g., ammonia ($NH_3$), nitrogen oxides ($NO_x$), nitrous oxide ($N_2O$), nitrate ($NO_3-$)). This is especially the case for non-point source Nr pollution from agriculture and rural areas[13]. Advanced technologies and management options reduce Nr losses from some forms, but may increase Nr losses of other forms[14,15]. Moreover, some abatement measures are rarely implemented to their full potential due to socioeconomic barriers, such as small farm size, which especially limits control of agricultural non-point source pollution in China[16,17]. It is therefore crucial to identify and realize synergies of Nr abatement across different sources and different nitrogen forms.

Urbanization is often associated with food security challenges as cropland conversion to urban areas reduces agricultural outputs and increases the concentration of pollutant emissions in population centres[18,19]. Unmanaged urbanization also holds many risks, such as inadequate water treatment infrastructure, leading to major pollution and health damages. However, research indicates that well-managed urbanization can benefit large-scale crop production since urbanization increased the total cropland area and the farm size which allows a faster introduction of modern agricultural practices, which potentially benefits crop-livestock integration for sustainable agriculture[20,21]. Meanwhile, urbanization moves rural populations to urban areas, which benefits the control of diffuse, non-point source pollution such as rural sewage, waste and livestock production.

Here we show the concept of well-managed urbanization pertains to the governance of land, resources, and the environment throughout the urbanization trajectory, deliberately omitting the management of social and economic aspects. It's worth noting, for example, that urbanization, when leveraged appropriately, has the potential to alleviate poverty and reduce inequality by expanding job prospects, improving living standards, enhancing educational opportunities, and bolstering healthcare services—factors that are outside the scope of our current investigation. Currently, approximately 14 million people move from rural to urban areas annually in China[22,23], which has an impact on the spatial distribution of resource use and environmental pollution along a rural-urban gradient. However, it has remained little understood how such a socioeconomic change affects Nr use and losses.

In this paper, we estimated the potential to halve nitrogen pollution in China through well-managed urbanization in all of the >2800 counties of China through analysis in four stages: 1) assessing changes of population, land use and agriculture with urbanization; 2) quantifying how these changes reduce Nr pollution from different sources and Nr forms; 3) estimating how these reductions in wasteful Nr losses on air and water quality; and 4) examining the feasibility and policy implications, including conducting a cost-benefit analysis.

## Results and discussion
### Changes with well-managed urbanization
Urbanization signifies a pivotal demographic transition from rural to urban living, a change that is profoundly shaping contemporary China. When effectively managed, urbanization can serve as a dynamic catalyst for various dimensions of sustainable development by conscientiously balancing environmental stewardship with socioeconomic considerations. This study, however, narrows its focus specifically to the aspects of the urbanization process that are intricately tied to nitrogen management through direct biophysical changes. This encompasses land intensification, the refinement of agricultural practices, and the expansion of treatment capabilities for domestic and industrial pollution.

Urbanization in China has been increasing by about 1% annually since the 1990s[22,23]. The United Nations' World Urbanization Prospects projected that China's urban population would increase from 0.8 billion in 2017 to 1.1 billion in 2050 with a population urbanization level of

around 80% if this trend continues (Fig. 1a, b and Fig. S1, S2). We predict that rural-to-urban migration also leads to a relocation of population from inland to coastal areas. Overall, population growth is projected to occur in the three largest coastal urban agglomerations, the Beijing-Tianjin region, the Yangtze River Delta and the Pearl River Delta, while the remaining regions would experience rural depopulation both through migration and natural aging (Fig. 1a). Rural depopulation and continued urban prosperity would lead to an increased level of urbanization in 82% of the counties, mainly in south of the Hu Line (Fig. 1e).

The expansion of urban area in response to urban population growth will take up natural and agricultural lands (Fig. 1c and Table S1). During the period from 2017 to 2050, China's total urban area is expected to increase from 5.5 to 7.8 Mha ( + 2.2 Mha, million hectares, an increase of about 40%), while rural built-up areas would decrease from 13.4 to 6.2 Mha (−7.2 Mha, ~−54%) (Fig. 1c, d). Although 1.3 Mha of cropland area would be occupied due to urban expansion, a total area of 6.9 Mha could be reclaimed from rural homesteads and converted to cropland. As a consequence, net cropland area can potentially increase by 5.6 Mha, mainly located in North China Plain and Northeast Plain (Fig. 1f). A decrease of cropland is estimated to mainly occur in the three largest metropolitan areas, Beijing-Tianjin region, the Yangtze River Delta and the Pearl River Delta.

The reclamation of rural land associated with urbanization could benefit the ongoing increase of large-scale farming practices, with opportunities to reduce nitrogen pollution and increase efficiencies. We estimate that the proportion of large-scale cropland farms (size over 10 ha) could increase from 9% (2017) to 90% in 2050 (Fig. S3). On the one hand, rural land reclamation increases the total area of croplands available; on the other hand, it contributes to increasing the spatial connectivity of cropland, since rural homesteads are normally next to croplands, which are naturally suitable to be reclaimed for better agricultural production.

Urbanization also drives the potential for coupled crop-livestock production. Smallholders normally quit livestock production due to having to keep part-time jobs in non-agricultural sectors to ensure their economic viability, which is not comparable with the time needed for livestock feeding[24]. In contrast, the rise of large-scale farming introduces more full-time professional farmers capable of managing both crop and livestock production simultaneously[25]. When accompanied by appropriate training and equipment, this enable crop-livestock coupling at the house hold level, as observed in areas adopting the new farming models (e.g., family farm, cooperative farm, industrial farm) with larger farm size and full-time farmers[26]. These new farming models would gradually replace smallholder farming as the dominant farming practice with less rural laborers and larger farm size by 2050, as has happened historically in other world regions. To ensure a well-managed transition, however, it is important that the nitrogen management opportunities are utilized, such as maximizing effective use of manures and other organic residues, while minimizing losses to the environment.

Additionally, the distribution of large-scale farms by 2050 would lead to livestock relocation while ensuring that livestock manure production matches crop requirements at the county level. Approximately 325 million pig units (all livestock species converted to pig units) would need relocation. For example, about 178 million pigs must move from the southern region (Fujian, Yunnan, Guangdong, Hunan, etc.) to the central region (Sichuan, Chongqing, Henan, etc.) (see Figs. 1g and S4).

In 2050, it is assuming that nitrogen management will adhere to the latest Five-year plan of the central government[27]. The government's plan involves collecting excreta from urban areas through a septic-sewer network, which will undergo treatment using nitrogen removal or recycling technologies[28] (Fig. S5). It aims to reduce Nr emissions by upgrading industries and restructuring energy consumption for

transportation and heating in urban areas. Managing non-point source pollution in rural areas presents a significant challenge, particularly with regard to wastewater and municipal solid waste. While controlling point source pollution in urban areas is comparatively more manageable, addressing non-point source pollution in rural areas is more complex. Therefore, the government has established precise indicators for wastewater and waste treatment efficiency in urban areas, and we assume that the status quo will persist in rural areas.

## Reduction of total Nr loss

In the well-managed urbanization scenario, national Nr losses would be reduced from 33.8 to 18.3 Tg (−46%), with reductions of 10.9, 3.4 and 1.2 Tg in agricultural, human and natural system losses, respectively

(Fig. 2). Agricultural systems, especially large-scale farming, full manure recycling, and emission abatement measures like reducing $NH_3$ emissions (Fig. 3a, b), drive the reductions. In agriculture, Nr inputs such as fertilizer, deposition, and irrigation decrease as professional farmers seek cost optimization, especially in a world with potentially higher nitrogen prices, while manure and straw recycling increases. In the human system, migration would increase the share of population connected to centralized sewage treatment (including nitrogen recovery opportunities) and foster household energy transformation, hence leading to both reductions of Nr emissions to water bodies and atmosphere in rural areas. The reduction of overall Nr losses to the environment from agricultural and human systems would result in lower Nr deposition and related Nr inputs, and thus also reduce Nr

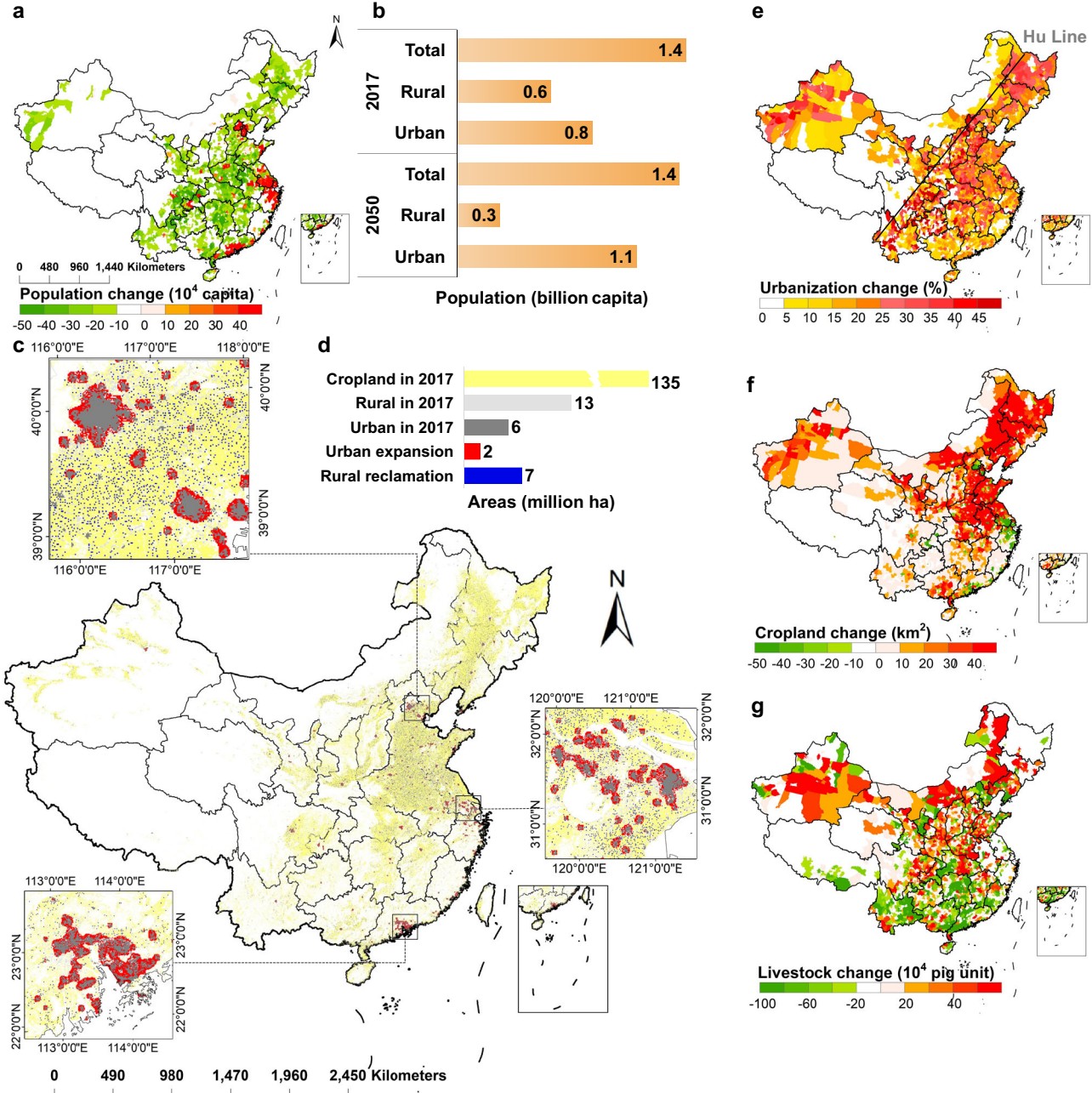

**Fig. 1 | Population, land use and agricultural change with urbanization.** These figures demonstrate changes in population, land use and agricultural system with urbanization from 2017 to 2050. **a** Geo-distribution of population change. **b** Change of population. **c** Geo-distribution of land use change, with the same colors as in **d**. **d** Quantity change of land use. **e** Geo-distribution of urbanization change. **f** Geo-distribution of cropland change. **g** Geo-distribution of livestock change. The base map is applied from GADM data (https://gadm.org/).

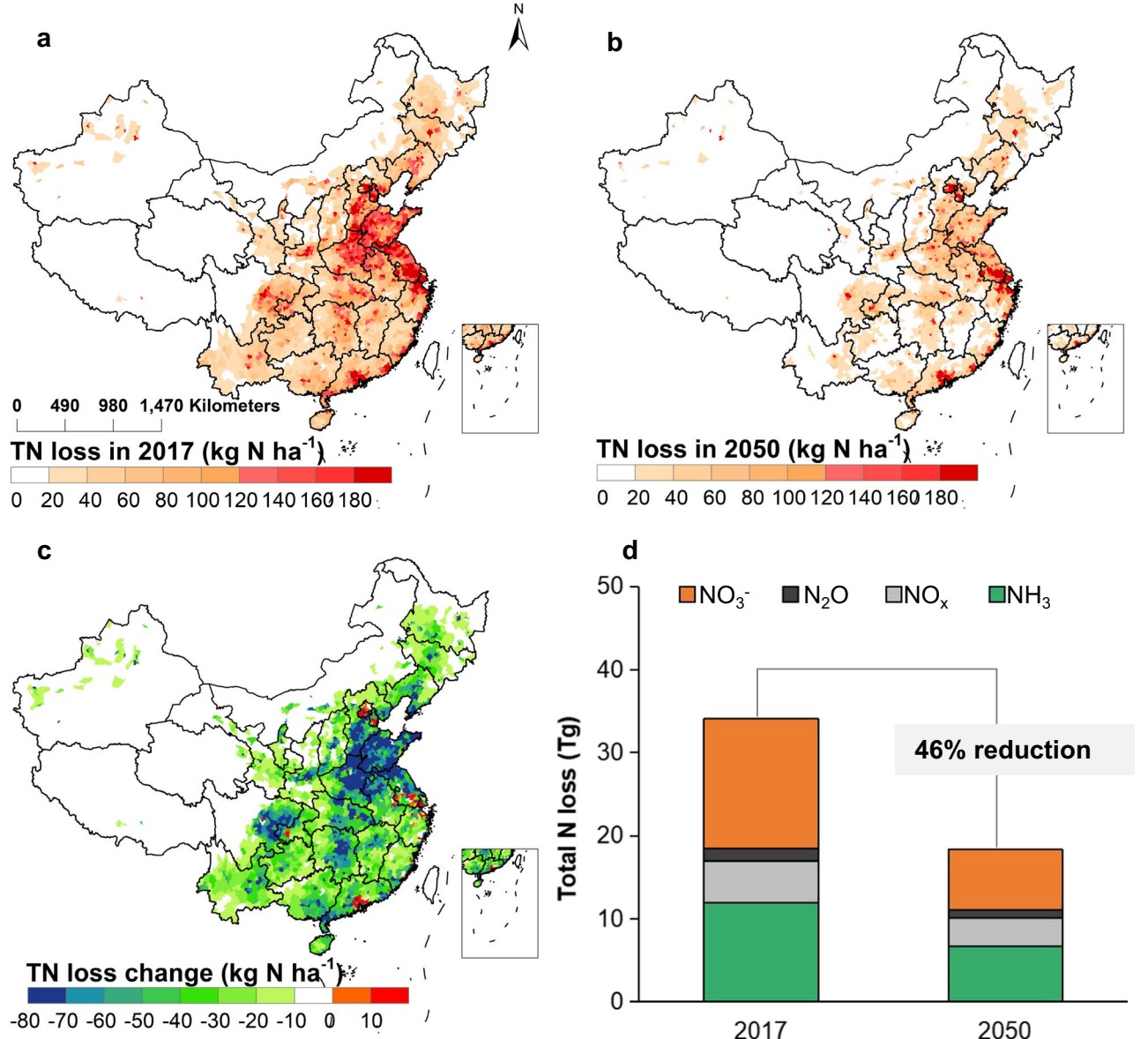

**Fig. 2 | Spatial variation of nitrogen loss with urbanization. a** Distribution of total reactive nitrogen (TN) losses in 2017. **b** Distribution of predicted TN losses in 2050 assuming an urbanization level of 80%. **c** Distribution of the change in TN losses during the 2017-2050 period. **d** National $NH_3$, $NO_x$, $N_2O$ and $NO_3^-$ loss in 2017 and 2050. TN losses include $NH_3$, $NO_x$, $N_2O$ and $NO_3^-$ loss. We use $NO_3^-$ to represent the Nr loss via runoff and leaching, given that most of the other dissolve forms of Nr would be converted to $NO_3^-$ during moving with water. The base map is applied from GADM data (https://gadm.org/).

losses from natural ecosystems, such as forests. Below we summarize the changes in nitrogen pollution according to the main forms of Nr loss to the environment:

$NH_3$ emissions are calculated to decrease from 11.9 Tg N to 6.7 Tg N (−5.2 Tg N, −44%), with 2.8 and 1.8 Tg N reductions from cropland and livestock systems, respectively (Fig. 3c). Such large reductions are mainly due to the enhanced Nr recycling ratio (NrR, which equaling manure nitrogen returned to field divided by the amount produced) and the increase in nitrogen use efficiency (NUE) of crop and livestock systems, the primary $NH_3$ source. Large-scale farming boosts cropland NUE from 43% to 54%, ultimately reducing total fertilizer use from 28.9 to 18.5 Tg N by 2050 (Fig. S6). Re-coupling crop and livestock production increases manure NRR from 23% to 63%, further reducing fertilizer use to 12.9 Tg N, with cropland NUE reaching 59%, similar to developed countries[29]. $NH_3$ emissions from human subsystems decrease by around 0.4 Tg N, primarily due to reduced emissions from rural sewage and upgraded industrial facilities in urban areas.

Improved management of sewage and industrial energy consumption in urban areas significantly contributes to reducing $NH_3$ emissions in non-agricultural sectors compared to free drainage systems in rural areas.

Well-managed urbanization is projected to reduce $NO_x$ emissions from 4.8 to 3.4 Tg N (−1.4 Tg N, −30%) (Fig. 3d), primarily driven by the human system (−1.4 Tg N). Fossil fuel consumption, responsible for over 80% of total emissions, is the main source. Well-managed urbanization would enhance energy efficiency, reducing $NO_x$ emissions per unit of energy supply, aided by growing urban populations and increased clean energy use in 2050. The Clean Air Act has reduced $NO_x$ emissions by 10% between 2015 and 2020 mainly through reductions per unit of energy supply[30], particularly in energy sectors, including rural bioenergy use[31]. Urban waste incineration is expected to rise, leading to an 0.1Tg N ( + 80%) increase in $NO_x$ emissions, but this only account for 7% of the total $NO_x$ emissions with reduced Nr runoff and leaching. Additionally, minor $NO_x$ emissions come from soil nitrogen

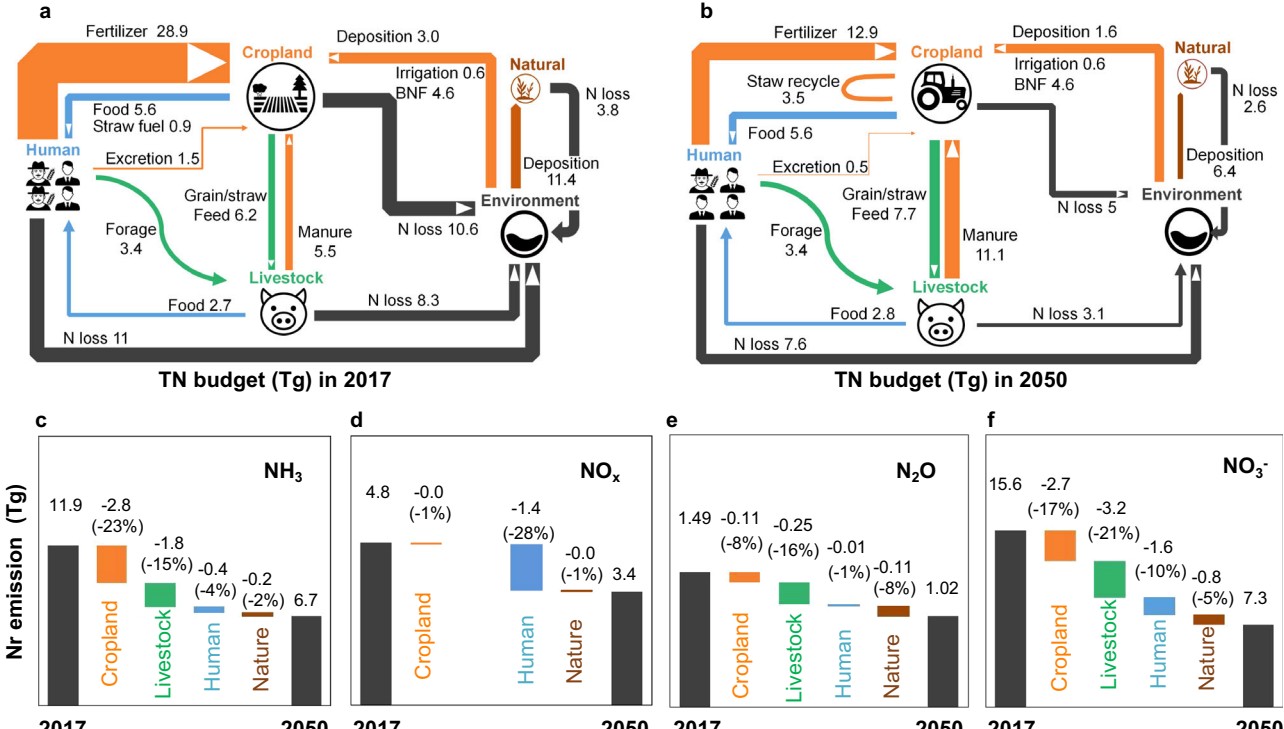

**Fig. 3 | Nitrogen budgets within key subsystems and their contribution to N losses reduction with urbanization.** Total N budgets for four subsystems and the environment in 2017 (**a**) and 2050 (**b**). These four subsystems, i.e. cropland, livestock, human and nature, are defined by the CHANS model. *Human* subsystem set includes urban human, rural human, wastewater treatment, garbage treatment and industry subsystems. *Nature* subsystem set includes grassland and forest subsystem. The colored arrows represent N flows to varying subsystems as follows: yellow, cropland subsystem; green, livestock subsystem; blue, human subsystem; brown, nature subsystem. The black arrows indicate the N flows that eventually enter the atmosphere and water bodies through gaseous emissions and runoff losses. **c**–**f** National nitrogen losses in 2017 and 2050, and the sources of the difference. The icons used in (**a**) and (**b**) are from https://uxwing.com/.

transformation, like denitrification. Reduced Nr input to agricultural and natural lands by professional farmers, including from Nr deposition, could reduce soil $NO_x$ emissions by 10%-30%.

$N_2O$ emission reductions, while relatively modest in absolute term (−0.47 Tg N), account for 33% of the total $N_2O$ emissions, making a significantly to climate change mitigation Fig. 3e). The largest contribution is from agricultural subsystem (−0.36 Tg N), where $N_2O$ mainly originates as direct emissions from cropland soil due to fertilization, nitrogen deposition and BNF, as well as manure recycling from livestock farming[4,32]. Even if cropland and livestock $N_2O$ emission factors remain constant, $N_2O$ emissions would be reduced due to reduced nitrogen input and increased manure recycling. Meanwhile, the reduction of nitrogen inputs in agricultural and human subsystems is calculated to reduce indirect $N_2O$ emissions by 0.11 Tg, which occur downstream or down-wind to forest and grassland via water bodies or the atmosphere[33].

Nr runoff and leaching to water bodies (dominated by $NO_3^-$, and labeled as $NO_3^-$ in Fig. 3, although also including other Nr forms including organic nitrogen) would be reduced from 15.6 to 7.3 Tg (−8.3 Tg N, −53%). The livestock subsystem leads this reduction (−3.2 Tg N) (Fig. 3f), through sustainable livestock management practices and livestock- cropland coupled system. The cropland subsystem contributes the second-largest reduction (−2.7 Tg N) due to reduced Nr inputs and increased NUE. In addition, Nr loss to water is predicted increase 0.5 Tg N in wastewater treatment plants, but decrease 2.1 Tg in wastewater directly discharge and garbage leaching. Domestic wastewater treatment capacity is increasing, aligning with China's 14th Five-Year Plan and 2035 Vision Plan[27,34], which emphasizes environmental investment to reduce Nr loss to water in urban areas[35].

Based on current knowledge, denitrification emissions of $N_2$ are highly uncertain but are still quantified in our approach for mass consistency as shown in Fig. S7. Although $N_2$ emissions are environmentally benign, they are usually ten times greater than $N_2O$ emissions and result in a significant loss of valuable Nr resources[36]. Reducing $N_2$ emissions and recovering associated Nr resources offers a cleaner, more circular system, minimizing fresh Nr inputs. Urbanization would lead to a 10% reduction in $N_2$ emissions by 2050 (−2.3 Tg N). Nitrogen removal technologies predict a 0.9 and 2.3 Tg N increase in $N_2$ generation in wastewater and garbage subsystems. Agricultural systems produce 40% less $N_2$ from denitrification (−3.8 Tg N) due to lower Nr input like fertilizer. Additionally, a 16% reduction in $N_2$ emissions (−1.0 Tg) results from reduced nitrogen deposition from natural subsystems.

While a few counties in megalopolises with populations exceeding 10 million, such as Beijing, Shanghai, and Shenzhen, are expected to have increased Nr losses to the environment due to urbanization by 2050, over 96% of counties are projected to witness reductions in Nr losses (Fig. 4). The North China Plain and Sichuan Basin stand out as key regions for reductions in $NH_3$, $N_2O$, and $NO_3^-$, driven by larger farms, livestock relocation, and improved sewage treatment (Fig. S8). While, The North China Plain is also a focal area for $NO_x$ reduction, primarily due to urban migration and industrial upgrades.

### Reduction of PM$_{2.5}$ and Nr output to water bodies

Curtailing nitrogen losses can have far-reaching consequences, including but not limited to air and water pollution, biodiversity diminution, and climate change. Within this spectrum of impacts, air pollution—predominantly attributed to PM$_{2.5}$, with a marginal contribution from ozone[11] —poses significant threats to human health.

Concurrently, water pollution and biodiversity depletion adversely affect ecosystem vitality. In light of these considerations, our following benefit assessment encompasses human health, ecosystems, and climate impact. Given that climate change is chiefly reflected in $N_2O$ emissions, our analysis delves deeper into $PM_{2.5}$ pollution and nitrogen losses affecting aquatic environments. $PM_{2.5}$ stands as a paramount

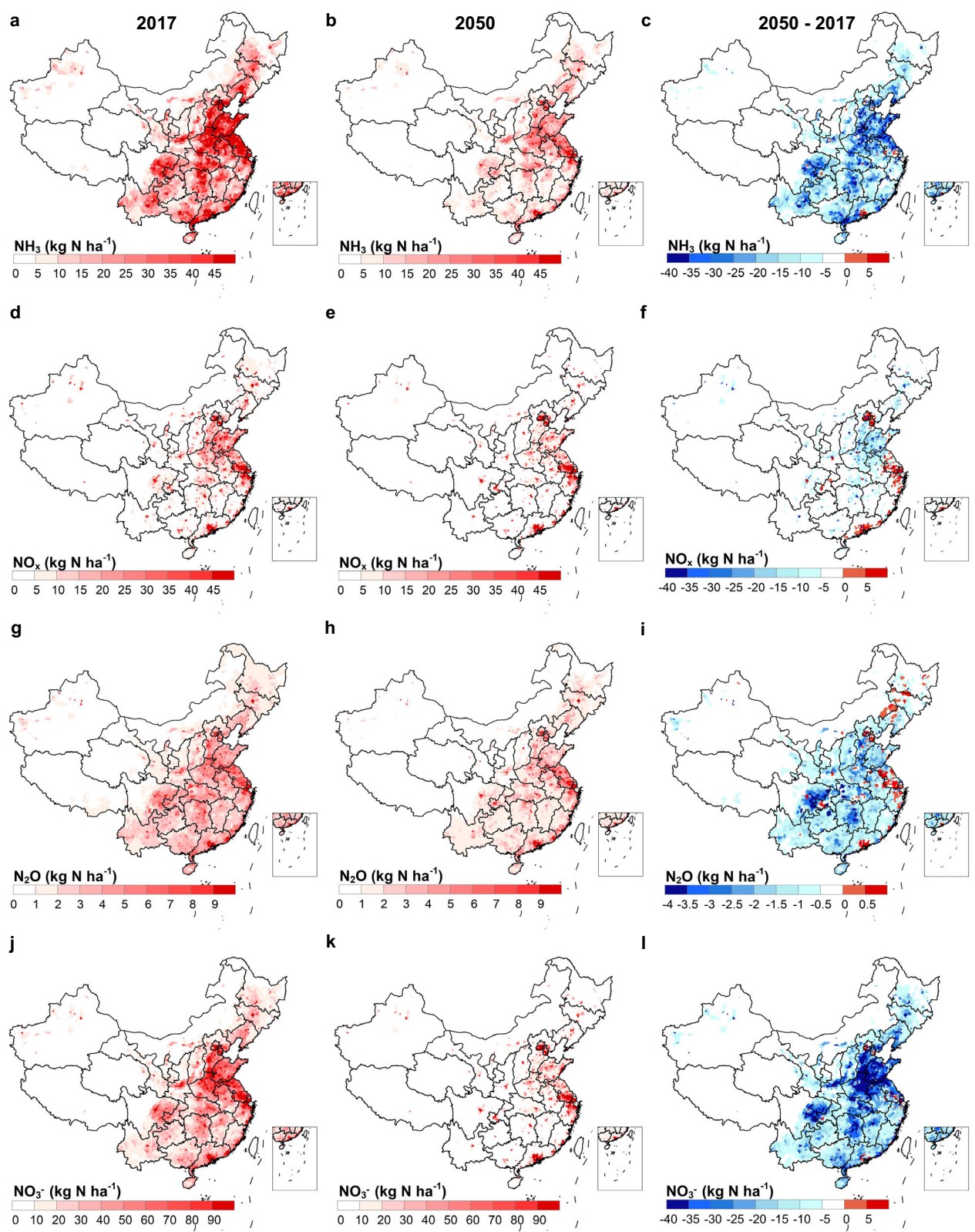

**Fig. 4 | Spatial variation of Nr species loss with urbanization.** Distribution of $NH_3$ in 2017 (**a**), 2050 (**b**) and the difference (**c**). (**d-f**), Distribution of $NO_x$ in 2017 (**d**), 2050 (**e**) and the difference (**f**). Distribution of $N_2O$ in 2017 (**g**), 2050 (**h**) and the difference (**i**). Distribution of runoff N loss in 2017 (**j**), 2050 (**k**) and the difference (**l**). The base map is applied from GADM data (https://gadm.org/).

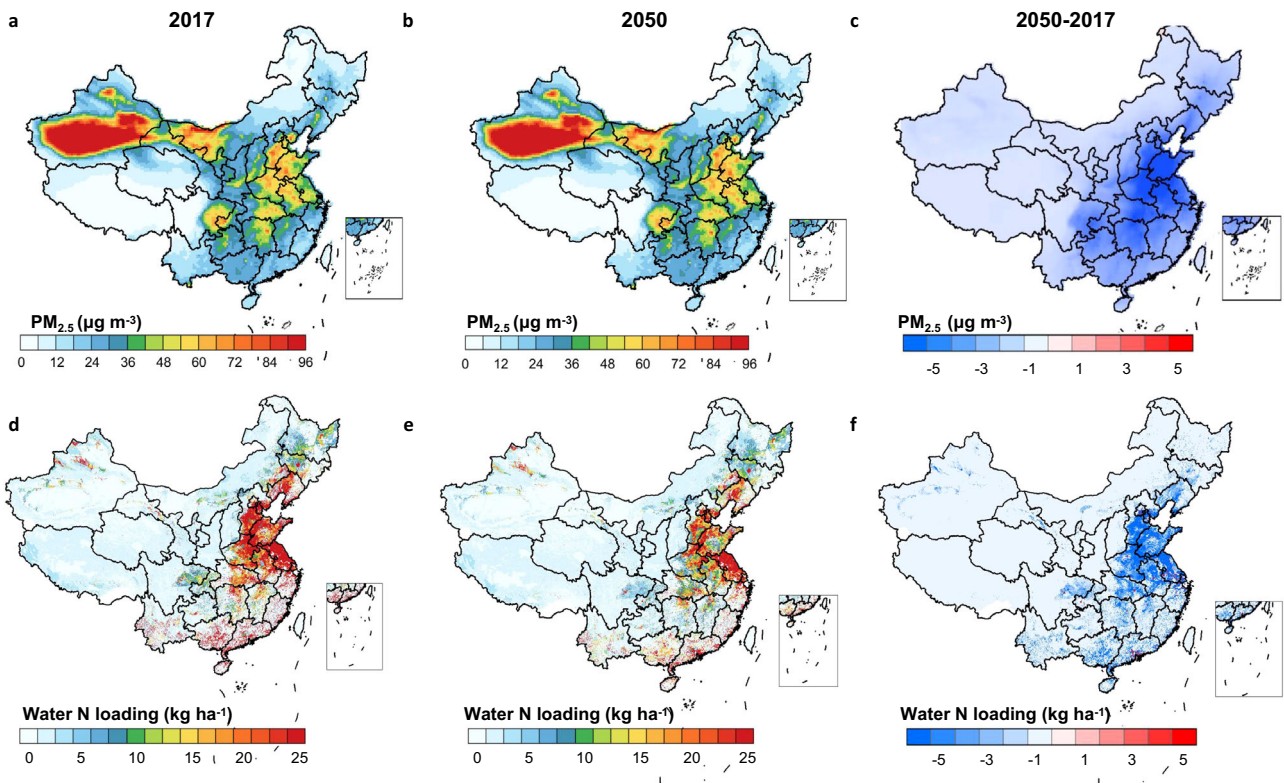

**Fig. 5 | Spatial variation of atmospheric PM$_{2.5}$ concentration and Nr to ocean with urbanization. a-c**, Geographic distribution of surface air PM$_{2.5}$ concentration in 2017 (**a**), 2050 (**b**), and the difference (**c**). Geographic distribution of Nr loading to downstream reach in 2017 (**d**), 2050 (**e**), and the difference (**f**). The base map is applied from GADM data (https://gadm.org/).

environmental precursor to premature mortality globally[37]. Our focus on the influence of adeptly administered urbanization on reducing PM$_{2.5}$ levels is predicated on the profound ramifications of PM$_{2.5}$-induced air pollution on human health, corroborating its global prominence as a mortality risk factor. Given the repercussions of PM$_{2.5}$ on human well-being, coupled with its significant presence in urban landscapes, this study emphasizes the shifts in PM$_{2.5}$ concentrations resultant from targeted reductions in Nr emissions.

Reduction of NH$_3$ and NO$_x$ emissions are predicted to decrease national average surface concentrations of PM$_{2.5}$ from 28.0 to 25.6 µg m$^{-3}$ (−2.4 µg m$^{-3}$, −9%). Regions with the largest reductions are projected to be in the North China Plain (Fig. 5a-c), and the largest reduction of PM$_{2.5}$ concentration could reach −7.7 µg m$^{-3}$, accounting for 18% of PM$_{2.5}$ concentrations in 2017.We predict that the three major coastal metropolitan areas, i.e., Beijing-Tianjin region, the Yangtze River Delta and the Pearl River Delta, would also experience reductions of PM$_{2.5}$ concentrations, although urbanization is likely to increase NH$_3$ and NO$_x$ emissions there. The meteorological conditions in these coastal areas, however, are more favorable for the dispersion of air pollutants, and more importantly, the large reduction of NH$_3$ and NO$_x$ emissions in surrounding regions will contribute to an overall reduction of PM$_{2.5}$ concentrations at regional scale. In line with the nationwide decrease of PM$_{2.5}$ concentrations, the national population-weighted PM$_{2.5}$ concentrations will drop from 41.4 to 32.8 µg m$^{-3}$ (−8.7 µg m$^{-3}$, −21%), and consequently reduce population exposed to the PM$_{2.5}$ levels above the national clean air standard (35 µg m$^{-3}$) from 609 million to 476 million (−22%)[38], leading to a marked public health benefit and reducing premature deaths.

Urbanization has also a strong reducing effect on Nr loading to water bodies and the related Nr export to the ocean from 6.8 to 3.5 Tg (−49%), especially in the Bohai Sea region, which has the potential to improve the eutrophication status of the coastal waters. Simulations based on the total Nr to surface water and purification capacity of lake, reservoirs, and rivers through denitrification and Nr accumulation in sediments. On the one hand, the rural-to-urban migration by 2050 is predicted to lead to a shift of hotspots of Nr losses from inland to coastal urban agglomerations, which weakens the purification potential of inland water bodies (−2%), due to the shortened retention time of Nr pollutants in the rivers. On the other hand, the reduction in domestic and agricultural Nr discharge is predicted to decrease the overall Nr loading to the lower reaches, with the most significant reduction occurring in the North China Plain (Fig. 5d-f). However, water Nr loading is predicted to increase in a few scattered reaches of rivers in the Pearl River Delta and the Yangtze River Delta.

## Cost-benefit assessment of Nr pollution abatement

The above-mentioned Nr pollution reduction potentials implies substantial benefits to the whole society, but it also requires the implementation of abatement measures with related costs. We calculate that approximate US$ 61 billion yr$^{-1}$ is be required to achieve the above-mentioned Nr pollution reduction potential (Fig. 6h). The one-time implementation cost of US$ 1404 billion are required for agricultural management optimization over the period of 2017-2050, including rural homestead reclamation, cropland consolidation and livestock relocation (Fig. S9). Assuming that these changes would be completed by 2050, the annual implementation cost would be US$ 43 billion. The cost of cropland consolidation is estimated around US$ 15 billion yr$^{-1}$, mainly in the North China Plain and the Middle-lower Yangtze River regions (Fig. 6b), which have the largest potential for the implementation of large-scale farming. As a result of the increase of large-scale farming, livestock production would relocate to these regions, with an annual cost of US$ 14 billion. The implementation cost of rural reclamation is estimated at US$ 13 billion yr$^{-1}$, more than half of which is calculated for the Beijing-Tianjin-Hebei region, Yangtze River Delta

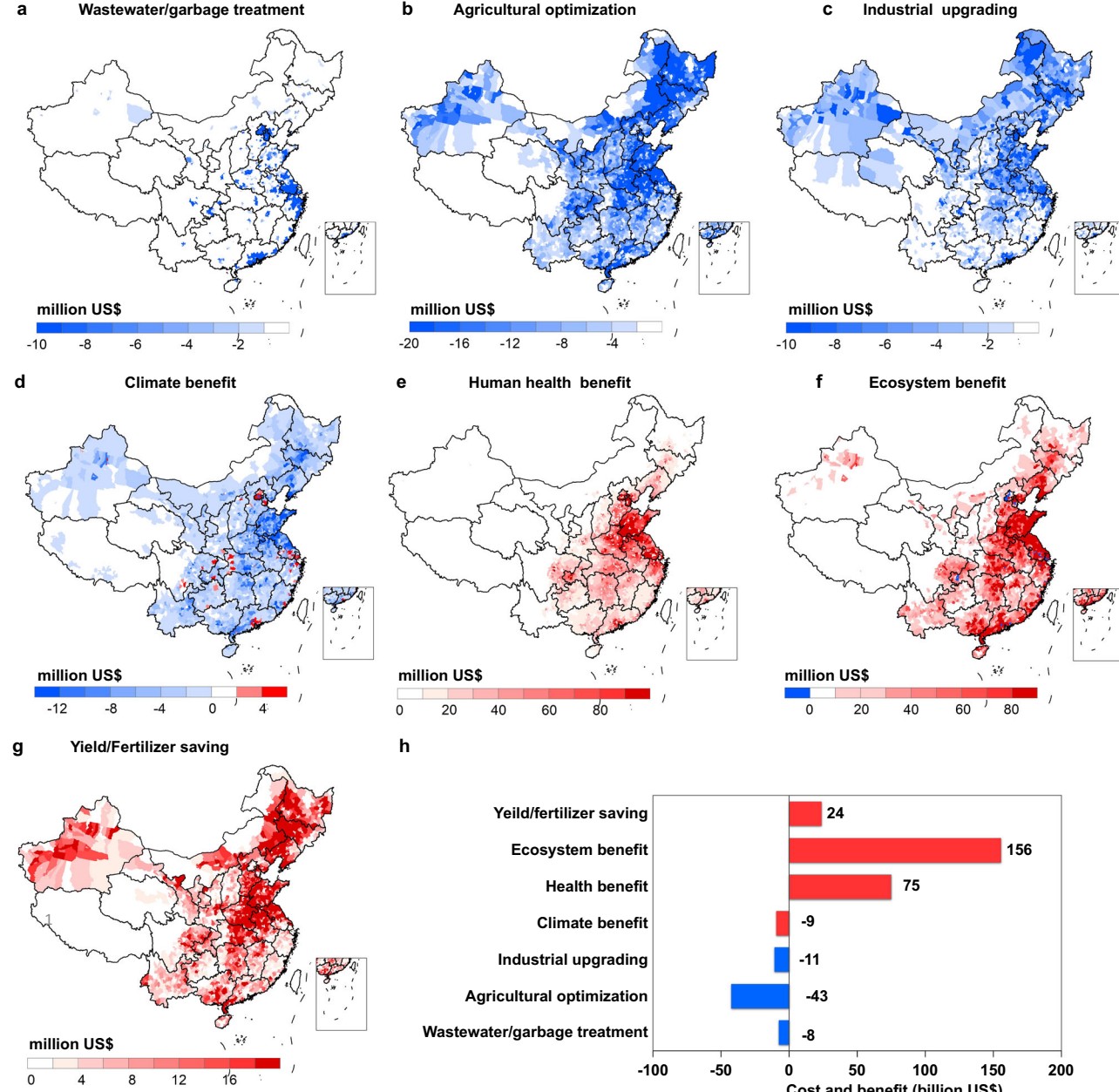

**Fig. 6 | Costs and benefits of halving Nr pollution with urbanization.**
**c** Geographic distribution of implementation cost on waste treatment (**a**), agricultural optimization (**b**), and industrial pollution system upgrades (**c**) to achieve the above mentioned reduction potential of Nr pollution with urbanization. Geographic distribution of monetized benefits on climate (**d**), human health (**e**) and ecosystem (**f**). **g** Geographic distribution of economic benefit from fertilizer saving and yield increase. **h** National costs and benefits with urbanization. Blue indicates implementation costs, while red indicates benefits. The base map is applied from GADM data (https://gadm.org/).

and Pearl River Delta. Although the rural reclamation area is small in these three regions, the unit cost per area is one order of magnitude higher than that in other regions due to their relatively higher economic development and land values.

In contrast to agricultural measures, the upgrading industrial processes and domestic waste treatment have lower implementation cost, requiring only US\$ 11 and US\$ 8 billion yr$^{-1}$, respectively (Fig. 6h). Industrial process upgrading is mainly utilized for energy sectors, such as NO$_x$ emission reduction measures in power plants and vehicles. The implementation cost includes both fixed input and operation cost, with the largest abatement cost in the North China Plain and the northeast regions (Fig. 6c). Relocating population from rural to urban areas would require more facilities to treat domestic sewage and solid

waste. We found that urbanization would lead to an additional 40 million tons of wastewater and 0.3 million tons of solid waste daily in urban areas by 2050, requiring roughly a US\$ 106 billion one-time investment in facilities construction. Assuming these facilities would be completed by 2050, the annual cost would be calculated at US\$ 3 billion. Beyond the construction of treatment facilities, the unit cost per ton to treat wastewater and solid waste is estimated at US\$ 0.16 and US\$ 2.8, respectively, leading to around US\$ 4 billion yr$^{-1}$ operating costs. The Beijing-Tianjin-Hebei region, Yangtze River Delta and Pearl River Delta are calculated to account for more than 60% of the total implementation cost for domestic waste treatment (Fig. 6a).

In contrast to these implementation costs, estimated benefits due to reduced regional Nr pollution, due to reduced impacts on human

and ecosystem health, climate benefits and increased economic returns through fertilizer saving and higher yields, and are estimated near US$ 245 billion yr⁻¹. Ecosystem benefits are estimated at US$ 156 billion yr⁻¹, mainly from the mitigation of eutrophication, improvement of drinking water quality and aquaculture production due to reduction of $NO_3^-$ loading of water bodies, as well as preserving forest biodiversity through reduction Nr emission and deposition. These benefits mainly occur in the Yangtze River Delta and Pearl River Delta regions where the largest reductions of $NO_3^-$ discharge are estimated and which have advanced levels of economic development (Fig. 6f). Reduction of $NH_3$ and $NO_x$ emissions will improve air quality due to the reduction of $PM_{2.5}$ and near-surface $O_3$ concentrations, with benefits to human health estimated at US$ 75 billion yr⁻¹, with the North China Plain being experiencing the most propound air quality improvements (Fig. 6e). Reduction of agricultural inputs from fertilizers is calculated to lead to savings of US$ 12 billion yr⁻¹, which would increase farmers' income. Similarly, increases in crop yields are predicted to generate US$ 12 billion yr⁻¹ economic returns to farmers. The reduction of $N_2O$ emissions would results in climate benefits valued at US$ 6 billion yr⁻¹, but being more than offset by the reduction of carbon sequestration (US$ −15 billion yr⁻¹) due to reduced Nr deposition, resulting in a net climate change effect valued at US$ −9 billion yr⁻¹. In total, urbanization could thus result in a societal benefit of approximately US$ 245 billion yr⁻¹ in China, approximately five times the total implementation cost, suggesting it would be both feasible and cost-beneficial to adopt ambitious targets for mitigating Nr. The North China Plain and the Middle-lower Yangtze River regions have the largest total benefit share and benefit-to-cost ratio, highlighting those as preferential areas for Nr management.

## Challenges and policy implications

This study introduces a series of measures to achieve effective resource and environmental management during urbanization process, alongside the goal of halving nitrogen pollution. However, the implementation of these measures presents institutional and socio-economic challenges. Firstly, the fragmentation of rural land ownership in China inhibits the development of large-scale farming, while land consolidation projects and livestock relocation require significant financial investments, with relatively low agricultural production benefits and weak incentives for producers. Additionally, despite the potential for high economic returns in urban industries to drive industrial upgrading and pollution control, this also necessitates standardized and regulated systems. Below, we will analyze the opportunities and feasibility of theses measure and how improve policy to address these challenges.

The Chinese government has promoted the scale of cropland tenure and the reclamation of homesteads through rural land system reform. With urbanization and rural aging, an increasing number of farmers voluntarily leasing their cropland to new farming models, which has reached 37 Mha as of 2019[39]. Simultaneously, the hollowing of villages has become widespread in central and western China as the younger generation has moved to urban areas. In this context, over the past decade, the potential rural reclamation area has exceeded 4000 ha[40]. However, additional efforts will be needed to address potential challenges such as land quality monitoring and maintenance of supporting infrastructure in response to potential land quality degradation resulting from the separation of land ownership and management rights.

An annual investment of approximately US$21 billion has been dedicated to transforming fragmented croplands into large-scale, high-standard farms. The Chinese government has already initiated a cropland consolidation initiative with the ambitious goal of achieving 72 Mha of high-standard croplands equipped with modern management and facilities in the forthcoming decades[41]. These efforts align with our urbanization projections and underscore the importance of conducting suitability assessments for high-standard farmland construction while avoiding high-cost, non-scalable areas. To enhance this endeavor, the government should refine the financing model by utilizing land tenure as collateral in land reform and increasing subsidized interest rates for high-standard farmland construction. Collaboration with institutions like the Agricultural Development Bank can further stimulate investments in cropland.

For livestock relocation, the Chinese central government requires newly-built livestock farms to be surrounded by a certain area of croplands for manure recycling. This ensures livestock relocation is not required. Where livestock farms have already been built, livestock relocation costs could be subsidized by government. It should be noted that cross-province relocation of livestock only accounts for 8% of livestock needed to be relocated (Fig. S4), suggesting the bulk of the relocation is estimated to occur within provinces, increasing the feasibility of livestock relocation due to farmers may not want to move far away from home. With the increase of large-scale croplands, having both crop and livestock production located close to each other becomes more viable for rural households, further aided by a reduction of implementation barriers by governments. In any case, subsidies supporting the relocation of livestock are vital, and central government has recently provided subsidies worth about US$ 750 million per county to improve the coupling of livestock and crop production as a measure to reduce environmental pollution originating from manure management in 100 demonstration counties[42].

Upgrading industrial and waste treatment facilities are essential measures responding to an increase in urban population. A continued focus on the implementation of the Clean Air Act and the Clean Water Act would further aid the reduction of Nr losses from the human system. Meanwhile, the newly implemented policy "Co-reduction of pollution and carbon emission" would link efforts for pollution control with the target of carbon neutrality (NetZero) while pursuing urbanization objectives. This will place a focus on measures which can achieve both goals in a cost-effective and overall cost-beneficial manner. These policy goals have already put Nr reductions on a good pathway, enhancing the feasibility of halving Nr pollution in the context of urbanization. Further actions to include measures and policies with relevance for Nr pollution reduction into wider considerations, including linking to climate change mitigation and NetZero targets, would contribute to achieving a range of SDGs in China and beyond.

Agricultural practices, urban growth, and urbanization management strategies in China exhibit considerable regional diversity. Regions such as the North China Plain, the Middle and Lower Yangtze River Plain, and the Sichuan Basin are anticipated hotspots of swift urbanization and are crucial food-producing areas. As of 2017, these areas primarily relied on small-scale farming characterized by excessive nitrogen fertilizer usage and suboptimal NUE levels of 20%-40%[43]. Additionally, these regions are known for monogastric animal farming, leading to lower NUEs compared to northern areas where ruminant farming prevails[44]. These specific regions are instrumental in curtailing nitrogen pollution, including $NH_3$ and $N_2O$ emissions as well as runoff losses. They present the most favorable benefit-to-cost ratio for intervention measures, underlining the urgency for expeditious land system reforms, upscaling to large-scale production, and embracing integrated crop-livestock methodologies.

Eastern China, in the throes of accelerated urbanization and industrial expansion, is critical in the battle against nitrogen pollution, especially concerning $NO_x$ emissions. The burgeoning populations in coastal urban conglomerates, notably the Beijing-Tianjin-Hebei cluster, predict a persistent surge in pollution emissions. In spite of the overarching developmental gains, this area is bracing for the urbanization-induced pollution repercussions. It is imperative for the region to steadfastly enhance domestic pollution treatment capabilities and enforce stringent emission controls, potentially through industrial evolution and structural overhauls.

Conversely, the Northwestern and Northeastern regions, despite their sparser populations, have triumphed in establishing extensive agricultural production, achieving NUEs above 50%—a feat unmatched by other regions in China[43]. With population dips and diminished urban stressors on the horizon, these territories require sustained land reforms and agricultural fine-tuning, although their prospects for nitrogen pollution mitigation are relatively constrained.

In the undulating terrains of Southwestern China, urbanization rates are slated to climb notwithstanding an overarching population contraction. This terrain, representing 10% of the country's arable land, faces challenges in transitioning to large-scale farming by 2050 due to its topographical constraints (Fig. S3b). Compounding these challenges are the poorer soil conditions relative to other regions. It's crucial to recognize that expanding agricultural landholdings here may inadvertently intensify soil erosion and nitrogen depletion[45]. To bolster nitrogen utilization efficiency and fortify food security, innovative strategies—such as altering crop compositions or strategic relocation of cultivable land—are worth investigating. Consequently, under existing policies, the scope for curtailing nitrogen emissions in these Southwestern highlands is circumscribed.

We propose nuanced urbanization management strategies tailored to China's diverse sub-regions, advocating for a unified national framework that endorses meticulously coordinated and well-managed urbanization. This strategic blueprint, emphasizing agricultural refinement, domestic pollution abatement, and industrial modernization, holds immense promise for propelling socio-ecological progress and realizing sustainable development ambitions.

## Limitation

Urbanization can affect dietary and production structures, which in turn can affect the use and loss of nitrogen. On the one hand, the increasing proportion of meat consumption due to income increase of urban residents can increase the production of livestock farming and the cultivation of animal feed such as soybeans. China has launched forage/soybean revitalization plans to address this issue. The increase in livestock farming can lead to more nitrogen loss, but the replacement of food crops with forage/soybean cultivation can increase NUE and reduce nitrogen loss[44]. In addition, the demand for aquatic products, which accounted for about 4% of China's food intake in 2021, is increasing, leading to an increase in aquaculture production. Generally, economic growth increases the demand for food and the use of nitrogen fertilizers, leading to more nitrogen pollution. However, at the same level of economic development, urban residents consume less food per capita than rural residents[46]. This complex process makes it difficult to predict how dietary changes will affect nitrogen cycling in urbanization in China. Moreover, this paper is more focused on well-managed urbanization that is defined at the beginning for only biophysical changes other than social and economic aspects. Thus, the extended implications of urbanization on future agricultural production structures and nitrogen pollution via dietary structure changes are beyond the scope of our present study and will be explored in future research.

We concur that the health ramifications of Nr emissions extend beyond diseases related to PM$_{2.5}$, encompassing ailments arising from such as the toxicity of NO$_x$ compounds. Our choice to use PM$_{2.5}$ concentrations as a quantifiable indicator of health benefits stems from the impact it has on human health and the availability of comprehensive assessment data. PM$_{2.5}$ is universally acknowledged for its severe and far-reaching health consequences. The health effects of PM$_{2.5}$ air pollution, which include conditions such as stroke, heart disease, lung cancer, and a range of chronic and acute respiratory diseases, have been meticulously evaluated. These evaluations are employed by the World Health Organization (WHO) for calculating the financial toll of air pollution-related health issues. While we recognize the value in assessing the toxicity of NO$_x$ for a more nuanced understanding of

health benefits, we faced modeling and methodological limitations that precluded us from distinctly parsing out the unique toxic effects attributable to NO$_x$ and other potential constrains.

This study utilized a sophisticated multi-modeling methodology to evaluate both air and water pollution. In particular, the PM$_{2.5}$ concentrations appraised for the year 2017 demonstrated a strong alignment with the actual measured values, as evidenced by a correlation coefficient ranging from 0.54 to 0.73 (Fig. S10). However, the validation of national-scale water quality models remains a complex endeavor, largely due to the limited availability of comprehensive time series data and information across various river sections. Consequently, we juxtaposed Nr water pollution levels with findings from preceding studies. Our analysis inferred that approximately 15.6 Tg of Nr was deposited into aquatic systems, a figure calculated through the CHANS model. This estimate closely mirrors the findings presented by the IMAGE-GNM (15.4 Tg)[47] and DNDC models (14.5 Tg)[10]. Moreover, our projection of Nr discharge into the ocean, derived using the WNF model, was approximately 6.8 Tg. This figure is conservative when compared to the 11.7 Tg suggested by the IMAGE-GNM model but exceeds the estimates put forth by the MARINA model (0.83-1.18 Tg)[48] and the official report from China, which stands at 2.3 Tg[49]. Despite the inherent uncertainties surrounding the extent of Nr contributions to marine environments, we ascertain our estimation outcomes to be within reasonable bounds. It's pivotal to note that this research primarily aims to discern the effects of proficiently orchestrated urbanization on nitrogen flows into rivers and oceans, rather than to quantify nitrogen loss on a national scale. The synergistic application of CHANS and WNF models ensures a harmonious simulation of nitrogen transit processes, implying that variations in nitrogen inputs into watercourses and seas are solely attributable to shifts in emission origins consequent to well-managed urbanization.

## Methods
### Data sources

High Resolution Remote Sensing Monitoring of Chinese Land Cover 2018 (HRRSM-CLC2018, 30 m) was derived from the Resource and Environment Data Cloud Platform (REDCP, https://www.resdc.cn/), which was used to model land use change with urbanization, as well as accompanying potential of large-scale cropland. The digital elevation model (DEM) data used for slope and elevation extraction was derived from Geospatial cloud (http://www.gscloud.cn/). Records of urban and rural population were derived from county-level census data set in China County Yearbook 2018, and provincial level statistical data from National Bureau of Statistics of China 2018 (http://www.stats.gov.cn/tjsj/ndsj/2018/indexch.htm). Records of livestock were derived from the Agriculture Organization's online statistical database (FAOSTAT, https://www.fao.org/faostat/en/#data/EMN), National Bureau of Statistics of China 2018 (NBSC, https://data.stats.gov.cn/easyquery.htm?cn=C01&zb=A0D0O&sj=2018) and Agricultural Pollution Source Census of China in 2017 (APSCC2017)[25]. The FAOSTAT and NBSC provides comprehensive survey data for livestock species in 2017. The APSCC2017 provides grided data of livestock numbers, production and excretion to cropland.

### Quantification of changes in urban and rural areas, farm size and crop-livestock recoupling

Changes in urban and rural area by migration with related changes in urban areas and crop land areas were simulated from 2017 onwards using the High-Resolution Remote Sensing Monitoring of Chinese Land Cover 2018 (HRRSM-CLC2018) at a 30 m x 30 m resolution as the base line. Migration and land use change modeling were based on following assumptions: (1) urban expansion has the highest priority, and can occupy surrounding grids, including cropland; (2) rural land reclamation is used to supply cropland; (3) no limits on migration distance. Spatial land use data and county level urban/rural population

data were transformed to gridded data. The change of urban and rural grids at a resolution of 30 ×30 m from 2017-2050 was predicted by the ArcGIS platform (version 10.6), aligning the number of people on urban and rural grids with the urban and rural populations in 2050 as projected by the United Nations World Urbanization Prospects (UNWUP, https://population.un.org/wup/Download/), respectively. More details of the prediction of urban expansion and land reclamation are in Supplementary Methods. Urbanization level of the county (*UL*) in 2050 was defined as the ratio of urban residents to the total residents in the county:

$$UL_t^{2050} = \frac{UP_t^{2017} + \Delta UP_t}{UP_t^{2050} + \Delta UP_t + RP_t^{2017} - \Delta RP_t} \tag{1}$$

where $UP_t$ and $RP_t$ are the urban and rural residents in the $t$th county, respectively. $\Delta UP_t$ and $\Delta UP_t$ are the changes on urban and rural residents during 2017-2050, respectively.

## Potential for Large-scale farming

Potential large-scale cropland is predicted and mapped on county scale, assuming it could be connected to the maximum extent, as long as not divided by other land uses, such as rivers, roads, villages, etc. The simulation method for potential large-scale farming by ArcGIS is described in detail in the Supplementary Methods. The planting structure of crops in this process remains consistent with year 2017.

## Crop-livestock coupling

Livestock was relocated based on the distribution of large-scale farming by spatial analysis and linear optimization. FAOSTAT and NBSC were used to estimate the livestock numbers. Details of conversion of pig units are shown in the Table S2, Fig. S4 and Supplementary Methods. The relocation process only takes into account livestock from agricultural areas, and the livestock production structure remains consistent with the year 2017. Livestock relocation potential ($LRPi$) is used to determine if the livestock in the $i$th grid exceeds the demand of the crop or the carrying capacity of the arable land, and to quantify the potential for moving out or in of livestock.

$$LRP_i = LCP_{max} - \frac{LI_i}{LCA_i}, if.LRP_{max} \le DN_i 2$$

$$LRP_i = DN_i - \frac{LI_i}{LCA_i}, if.LRP_{max} > DN_i \tag{3}$$

$$DN_i = F_t^{2050} + M_i \tag{4}$$

where $LCP_{max}$ represents the maximum manure Nr carrying capacity of cropland (around 144 kg N ha$^{-1}$)[50], $LI_i$ represents the total manure Nr (kg) in the county where the $i$th grid is located (kg), $LCA_i$ represents the areas of large-scale cropland where the $i$th grid is located (km$^2$), $DN_i$ represents the crop demand for Nr, calculated as the sum of optimal fertilizer application rates ($F_i$, kg ha$^{-1}$) and manure Nr returned to cropland ($M_i$, kg ha$^{-1}$) in 2050. We assume that $F_i$ equals the synthetic fertilizer inputs after cropland scale-up, as estimated in Eq. (7).

Given reducing the cost of demolition and construction, the livestock would be prioritized to move from the N overload grids into the surrounding grids. Statistics Tool in ArcGIS was used to calculate the priority of all non-overload grid ($LP_i$) with a value of 1 to the overload grid ($LRPi \le 0$) and 0 to the other grid ($LRPi > 0$), respectively.

$$LP_i = \frac{N_i}{20 \times 20} \tag{5}$$

where $N_i$ is the number of overloads grids within surrounding 400 (20×20) grids and $i$th grids in the central of 400 grids. Based on the following linear programming, grids are selected for livestock moved in.

$$maxmize\, n$$

$$s.t \sum_{j=1}^{n} LRP_{j,r} \le PLRP_r \tag{6}$$

where $j$ is the serial number for non-overload girds in descending order according to $LRP_{j,r}$, $r$ represents province, $PLRP_r$ is the livestock relocation capacity in the $r$th province. If the provincial livestock volume exceeds the provincial carrying capacity, it is assumed that the maximum livestock volume is matched on all the large-scale cropland. The excess livestock volume is moved to the nearest province, and the method of spatial allocation is the same as the intra-province relocation. Livestock cannot be relocated to provinces with restricted development based on the vulnerability level of the surface water, as determined by the Ministry of Agriculture and Rural Affair.

According to the National Key Research and Development Program in China, we assume that crop-livestock coupled would reduce 60% NH$_3$ in manure management[51]. This reduction can be achieved through various measures, including slurry acidification, covering slurry storages, and using closed manure composting technologies. Additionally, when manure is recycled to replace synthetic fertilizer, the NH$_3$ emission factor is assumed the same. Furthermore, due to strict water pollution regulations, direct discharge of NO$_3^-$ is assumed not allowed in 2050[51]. Table S3 and Table S4 provides further details on the optimization of various parameters.

## Quantification of changes in urbanization on Nr pollution

Large-scale farming can increase the adoption of new technologies such as 4 R nutrient stewardship (i.e., the right rate, type, time, and place of fertilizer application), optimal irrigation techniques, and organic amendments, resulting in reduced fertilizer application and improved nitrogen use efficiency. This, in turn, can reduce fertilizer NH$_3$, N$_2$O and N loss to water. The estimation of influence of farm size on fertilizer use and NH$_3$ emission followed Duan et al., 2021[52] and Wang et al., 2022[53]. The Eqs. (7) and (8) are used to calculate fertilizer use and NH$_3$ emission on county scale in 2050.

$$F_t^{2050} = \exp\left(\alpha \times LnX_t^{2017} - \alpha \times LnX_t^{2050}\right) \times F_t^{2017} \tag{7}$$

$$E_t^{2050} = \exp\left(\beta \times LnX_t^{2017} - \beta \times LnX_t^{2050}\right) \times EF_t^{2017} \times F_t^{2050} \tag{8}$$

where $F_t$ represents the synthetic fertilizer input in $t$th county (kg). $X_t$ stands for the farm size in $t$th county (ha). $E_t$ represents the total NH$_3$ emission in $t$th county (kg), $EF_t$ represents NH$_3$ emission factor in $t$th county (%), $\alpha$, $\beta$ refers to estimated coefficients of large-scale farming from Duan et al.[52] and Wang et al., 2022[53].

Meanwhile, the management of livestock would be optimized through mitigation measures, such as surface coverage, frequent removal of manure from animal houses, and low crude protein feeding to reduce ammonia volatilization. We cited the research findings of Zhang et al., 2020[54], which showed that a combination of different measures in three stages of feeding (including low-protein feed and phased feeding), housing (including ventilation systems, floor mats, dry manure removal, and solid-liquid separation), and manure management (including anaerobic digestion, and high-temperature composting) can achieve a 60% reduction in NH$_3$ emissions from the livestock farming process in China.

Since there is no direct report on the relationship between farm size and other Nr loss to water, the study assumed constant loss

efficiency runoff he leaching in the field. The reduction of agricultural nitrogen losses may be underestimated here. In addition, the adoption of a crop-livestock coupled system can ensure that all manure is collected and applied to cropland[51].

An inverted U-shaped relationship between urbanization and industrial pollutants in China was identified based on Stochastic Impacts by Regression on Population, Affluence, and Technology (STIRPAT) model and panel data[55,56].The changes of industrial $NO_x$ emission ($NE$) and $NO_3^-$ discharge ($ND$) with urbanization were estimated as follow:

$$NE_g^{2050} = \exp\left(\alpha \times LnNE_g^{2017} - \alpha \times LnNE_g^{2050}\right) \times NE_g^{2017}$$
$$ND_g^{2050} = \exp\left(\beta \times LnND_g^{2017} - \beta \times LnND_g^{2050}\right) \times ND_g^{2017} \quad (10)$$

$NE_g$ and $ND_g$ represent industrial $NO_x$ emission and $NO_3^-$ discharge in the $g$th province (kg) respectively, $\alpha$ and $\beta$ refers to estimated coefficients from Xu et al.[55,56]

### Nitrogen budget
The CHANS model[36,57] was used in this study to analyze the impact of urbanization on the national nitrogen budget (Fig. S1). The CHANS model has 14 subsystems (e.g., croplands, livestock, human, industry, forest, grassland, atmosphere, water bodies, etc.) that covers all the nitrogen budget calculations in this paper. The baseline of nitrogen budget in 2017 was built first, then urbanization scenarios with corresponding parameters adjusting and urban/rural subsystem developing integrated into CHANS model to forecast the nitrogen cycle and loss in 2050. Urban/rural population change and accompanying difference of domestic pollution treatment efficiency are the key parameters which affect nitrogen budgets through urbanization. Details about the data sources, parameters and modeling methods can be found in Table S3 and Table S4. Annual nitrogen emissions, encompassing $NH_3$, $NO_x$, $N_2O$, and $NO_3^-$, derived from each subsystem of the CHANS model, were allocated to counties using proxy parameters, relevant methodology is detailed in Supplementary Methods and Table S5.

### Quantification of reductions in Nr losses on air $PM_{2.5}$
Weather Research and Forecasting model coupled with Chemistry (WRF-Chem)[58] was used to estimate the changes on surface concentrations of $PM_{2.5}$ based on $NH_3$ and $NO_x$ emission reduction through well-managed urbanization. The baseline simulation in 2017 included all anthropogenic and natural emissions. Anthropogenic emissions data of air pollution was taken from the MEIC 2017 for mainland China[59,60] and the MIX Asia emission inventory for the rest of the WRF-Chem domain[61] (www.meicmodel.org). The $NH_3$ and $NO_x$ emission inventories were replaced with the results of this study, thus enabling the assessment of the impact of urbanization-reduced $NH_3$ and $NO_x$ emissions on $PM_{2.5}$. More details for model simulation and validation are given in the Supplementary Methods.

The population-weighted $PM_{2.5}$ concentration ($PWC$), which considers population as weights at different exposure to $PM_{2.5}$, is used to reflect the impact of $PM_{2.5}$ concentrations on the national population. It is defined as:

$$PWC = \frac{\sum_{i=1}^{n}(P_i \times C_i)}{P} \quad (11)$$

where $PWC$ is population-weighted $PM_{2.5}$ concentration in China, $P_i$ is the population in the $i$th grid, $n$ is the total number of grids in the China, $C_i$ is the $PM_{2.5}$ concentration in the $i$th grid, and $P$ is the total population of China.

### Quantification the reductions of Nr output to sea
A water network-based framework (WNF)[62] was used to estimate the changes on Nr output to sea based on Nr loss to surface water through well-managed urbanization. WNF is parsimonious solution for surface water modeling that incorporates topology structure, hydrological and biogeochemical processes. And it ensured that Nr transport processes were simulated in complete agreement and the loss of Nr to ocean is only influenced by the changes in N loss to surface water. Reactive nitrogen loss to surface runoff (SR) depend on the Nr input to the grid and the loss coefficient determined by the type of land use.

$$SL = \sum_{k=1}^{m} Input_k \times \alpha_k \quad (12)$$

where $Input_k$ represent Nr input from the $k$th source, covering all of anthropogenic and natural sources, i.e. cropland grid input includes fertilizer, deposition, irrigation, livestock manure, human excretion, cropland BNF and straw recycling.$\alpha_k$ is the surface runoff loss coefficient[36]. Details about Nr transfer process estimation during surface water based on WNF are given in the Supplementary Methods.

### Quantification of costs and benefits of urbanization
Nr related implementation cost with urbanization includes the costs of construction and operation of waste treatment facilities, as well as the costs of agricultural management optimization and industrial upgrading.

To assess unit cost of sewage/garbage facilities construction (CSCost and CGCost), over 200 actual investment projects for waste treatment were collected from the Public-Private Partnership Service Platform under the National Development and Reform Commission of China (www.chinappp.cn). All cost data from the literature or web were measured in constant 2017 US$ (i.e., 100 CNY = US$ 16.16). The numbers of projects and unit investment costs for each region are given in Table S6. In addition, the referenced daily unit costs of operating wastewater treatment plants, landfill plants and waste incineration plants (US$ 0.2, 6.5 and 23.0 $t^{-1}$) were used to calculate the operating costs[63-65]. Annual construction and operation costs of waste treatment facilities (CFC and OFC) were estimated on county scale as follows:

$$CFC = \sum_{t,n} \left[ (SD \times CSCost_t + GD \times CGCost_{t,n} \times R_n) \times MP_t \right]/YS \quad (13)$$

$$OFC = \sum_{t,n} \left[ (SD \times OSCost_t + GD \times OGCost_{t,n} \times R_n) \times MP_t \right] \times 365 \quad (14)$$

where $t$ represents >2800 counties, $SD$ and $GD$ represent the daily production of sewage and garbage per capita (0.15 t capita$^{-1}$ and 1.17 kg capita$^{-1}$, respectively)[66,67], $MP_t$ represents the numbers of migrated people in the $t$th county (million capita), $YS$ represents the time span from 2017 to 2050, equal to 33 years, CSCost and OSCost represent the unit cost for the construction and annual operation of sewage treatment facilities, respectively (US$ $t^{-1}$), CGCost and OGCost represent the unit cost for the construction and annual operation of garbage treatment facilities, respectively (US$ $t^{-1}$), $n$ represents two types of garbage disposal, i.e. incineration and landfill and $R_n$ is the predicted ratio of incineration and landfill (65% and 35%), referring to the national 14th Five year plan[27],

Agricultural management would be changed with urbanization in China, with the increasing on the size and scale of farmland and coupling crop-livestock production. Implementation cost of this agricultural optimization is calculated including homestead reclamation, cropland consolidation and livestock relocation. We did not factor in the cost of homeownership for farmers relocating to urban areas, as we

assume their migration is part of natural process of urbanization, not a well-managed measure, and their implementation costs have already been balanced through market regulations. Cost data of homestead reclamation and cropland consolidation were collected from more than 200 projects in China Land Consolidation and Rehabilitation (CLCR, http://www.lcrc.org.cn), and were divided into three regions according to migration and economic development levels (Table S7). Annual construction cost of rural reclamation and cropland consolidation (RRC and CCC) were estimated on county scale.

$$RRC = \sum_t RA_t \times RRCost_t / YS \qquad (15)$$

$$CCC = \sum_t CA_t \times CCCost_t / YS \qquad (16)$$

where $RA_t$ and $CA_t$ are the areas of rural reclamation and cropland consolidation in the $t$th county (ha), respectively. $YS$ represents the time span from 2017 to 2050 (33 years). $RRCost$ and $CCCost$ are the unit cost of rural reclamation and cropland consolidation (US\$ ha$^{-1}$), respectively.

The cost of livestock relocation ($LC$) mainly includes the costs of dismantling livestock farms and reconstruction new livestock farms. The cost of dismantling a livestock farm includes the cost of the farming facilities and the livestock in stock. Cost data on facilities and livestock (FLCost and LLCost) were obtained from around 35 local compensation programs by local governments for the removal of livestock farming (Table S7). Reconstruction cost (RCCost) was derived from National Compilation of Agricultural Cost-benefit information 2018[68].

$$LC = \sum_t \left[ PR_t \times (\varepsilon \times FLCost_t + LLCost_t)/YS \right] + \sum_t PI_t \times RCCost \qquad (17)$$

where $PR_t$ represents the livestock that need to be moved out of $t$th county (unit pig), $PI_t$ represents the livestock that need to be resettled in the $t$th county (unit pig), $\varepsilon$ represents the area of land occupied per unit pig (2.3 m$^2$ per pig)[69], $FLCost$ and $LLCost$ represents the unit cost for removal facilities and livestock (US\$), respectively. $RCCost$ represents the unit product cost of livestock (US\$)[68].

The implementation cost of industrial upgrading is defined as direct expenditure for the abatement measures with urbanization to reduce Nr emission. Here we mainly refer to the database and methodology of cost assessment from the online Greenhouse Gas and Air Pollution Interactions and Synergies (GAINS) model (https://gains.iiasa.ac.at/models/index.html) to calculate the abatement costs of industrial mitigating Nr emissions. International technology cost data have been taken into account and adjusted in GAINS for province-specific conditions in China, which take into account Consideration of local labor costs, energy prices, costs of by-products, etc. The costing module of GAINS has been described in detail by Klimont and Winiwarter[70] and Zhang[54]. The annual cost of implementation ($COST_{ind}$) is calculated as follows:

$$IC = \sum_{t,q} \Delta NR_{t,q} \times ICost_{t,q} \qquad (18)$$

where $q$ represents the forms of Nr losses including NH$_3$, N$_2$O, NO$_x$ and NO$_3^-$, $\Delta NR$ is the reduction of Nr through industrial upgrading in county scale, $ICost$ is the integrated mitigation unit cost of industrial upgrading to reduce Nr emission (US\$ per kg N) (Table S8), which is derived from the online GAINS model database.

The total benefit (TB) of reducing Nr loss with urbanization include human health benefit (HBq), ecosystem benefit (EBq) and climate benefit (CBq) and economic benefit (EcB) by reducing agricultural inputs and increasing total yield as follow:

$$TB = \sum_q \left( HB_q + EB_q + CB_q \right) + EcB \qquad (19)$$

where $q$ represents the forms of Nr losses, i.e. NH$_3$, NO$_x$, N$_2$O and NO$_3^-$. HB, EB, and CB denote social benefits, which need to be defined in conjunction to Nr loss and converted for monetization. The impact of Nr loss on human health is focused on PM$_{2.5}$ caused various disease, such as ischemic heart disease (IHD), lung cancer (LC) and so on. The impact on environment mainly refers to water pollution, i.e. eutrophication and biodiversity loss of aquatic ecosystems, due to NO$_3^-$ loss to water, as well as soil acidification and biodiversity loss of terrestrial ecosystems due to Nr emission and deposition. The impact of Nr loss on climate change is complex, with benefits from N$_2$O reduction and damages from NH$_3$ and NO$_x$ reduction that affecting carbon sequestration in forests. The monetary benefit of HB, EB and CB were estimated by multiplying social benefits per unit Nr by reduction in Nr with the following equations:

$$HB_q = \Delta NR_q \times HCost_q \qquad (20)$$

$$EB_q = \Delta NR_q \times ECost_q \qquad (21)$$

$$CB_q = \Delta NR_q \times CCost_q \qquad (22)$$

where $\Delta NR_q$ is the reduction of $q$th Nr with urbanization in China (kg). $HCost_q$, $ECost_q$ and $Ccost_q$ are the unit damage cost of the three societal benefit due to the $q$th Nr loss (US\$ kg N$^{-1}$), respectively. $HCost_q$ was calculated by the cause-specific integrated exposure-response functions which can estimate the relative mortality risk from exposure to PM$_{2.5}$[11]. $ECost_q$ was derived from the nitrogen assessment in Europe after correction for difference in the willingness to pay (WTP) and the purchasing power parity (PPP) between EU and China[71,72]. $Ccost_q$ was calculated based on the marginal abatement cost (the carbon price) to reduce or increase emission of CO$_2$-eq[73]. A detailed method description of the unit damage cost of varying forms of N loss can be found in Table S9. HB, EB, and CB are assigned to counties based on relevant parameters, for example, changes in air and water pollution, population density, and VSL were used to correct for HB. More details are given in the Supplementary Methods.

Agricultural management optimization during urbanization increases economic benefit (EcB) by reducing agricultural inputs such as fertilizers, and increasing total yield due to cropland area increase.

$$EcB = \sum_t \left( A_t \times Y_t^{2017} \times PC + F_t \times FC \right) \qquad (23)$$

where $t$ represents >2800 counties; $A_t$ is the increasing potential of cropland due to homestead reclamation in the $t$th county (ha); $Y_t$ is crop yield in the $t$th county in 2017 (kg ha$^{-1}$), which is derived from the National Bureau of Statistics (https://www.stats.gov.cn); $PC$ is the average market price of the wheat in China in 2017 (US\$ 0.38 kg$^{-1}$), which was from National Development and Reform Commission (https://www.ndrc.gov.cn/xxgk/zcfb/tz/201610/t20161021_963246.html); $F_t$ represents the reduction of synthetic fertilizer input of $t$th county related to an increased NUE, assocoatied with an increased farm size, as given in Eq.7 (kg); $FC$ is the unit cost of Nr fertilizer (US\$ 0.66 kg N$^{-1}$), estimated with reference to the 2017 urea market price from the price monitoring network of the Ministry of Agriculture and Rural Affairs, PRC (http://zdscxx.moa.gov.cn:8080/nyb/pc/search.jsp).

## Data availability

Data supporting the findings of this study are available within the article and its supplementary information files.

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

## Acknowledgements

This study was supported by the National Natural Science Foundation of China (No. 42325707 to B.G.) and (No. 42261144001 to B.G.), and National Key Research and Development Project of China (No. 2022YFE0138200 to B.G.). This work is a contribution from Activity 1.4 to the 'Towards the International Nitrogen Management System' project (INMS, http://www.inms.international/) funded by the Global Environment Facility (GEF) through the United Nations Environment Programme (UNEP), and Frontiers Planet Prize Award: International Champion Prize funded by the Frontiers Research Foundation.

## Author contributions

B.G. conceived the study. O.D., J.R. and S.W. performed the urbanization modeling. J.D., Jian.X and S.H. performed the agricultural optimization modeling. O.D. and X.Z. performed the CHANS modeling and cost-benefit. L.Z., Y.X. and Z.X. performed the air and water quality estimation. O.D. and B.G. wrote the first draft, while S.R., Jiay.X. W.V and M.A.S revised the paper.

## Competing interests

The authors declare no competing interests.
