## [Peer Review File · Nature Communications]

Managing urban development could halve nitrogen pollution in ChinaREVIEWER COMMENTS

Reviewer #1 (Remarks to the Author):

Deng and colleagues present a comprehensive method coupling N loss prediction and cost-benefit analysis in order to assess the role of well-managed urbanization in N pollution mitigation. The work is solid and the manuscript is well organized. Nevertheless, there are a few issues which need to be addressed, I elaborate these aspects upon which my assessment is based.

(1) There are two threads presented in this study. The first part is the changes prediction and N loss reduction with well-managed urbanization. The second part is the cost-benefit analysis of N pollution abatement. They are strangely connected by air PM 2.5. The reduction of N loss could cause lots of impacts, such as the decrease of eutrophication, ozone damage, photo-chemical smog, health damage and other damage. Why only the PM 2.5 is highlighted?

(2) Along this line, in cost-benefit analysis, could the impact of N loss on human health be fully assessed by only focusing on PM2.5 caused various disease? How about the toxicity of NO_x and related damage to lung and central nervous system?

(3) The authors elaborate the changes with well-managed urbanization, including population migration, agricultural optimization and industrial upgrading. It seems to be a comprehensive optimization of all population, land and industry resources all over the country. Therefore, I wonder the definition or the boundary of well-managed urbanization. Whether all the changes you mentioned are driven by urbanization? Whether the urbanization can outline all the perspectives for the future social development?

(4) In the results part, the North China Plain is so conspicuous due to high N loss in 2017 and high reduction potential. However, in the Northeast area, where agricultural production holds large proportion in the total production nationwide, and also will experience intensive population reduction, urbanization increasing and cropland change according your prediction shown in Fig. 1, has still low TN loss potential (Fig. 2c). Why? More analysis and discussion for spatial heterogeneity of N loss is needed combining with actual industries, agriculture (such as coupled crop-livestock production), economy, population and policy, which could also provide implications for practice.

(5) I noticed that authors opted for including so-called "S2 Comparison of different water quality model" in supporting information. Including discussion item in this section seems not appropriate and therefore I invite the authors to critically evaluate whether this information could be shifted to the main document.

(6) Generally speaking the manuscript is well organized and formatting is mostly adequate. Nevertheless, I spotted several mistakes listed as follows. Therefore, I recommend the authors to re-check the whole manuscript.

Line 590, the reference is cited repeatedly.

Equation 9, line 588, I believe it should include a NO₃ discharge (ND) rather than all NE.

Reviewer #2 (Remarks to the Author):

The manuscript titled "Well-managed urbanization could halve nitrogen pollution in China" aims to investigate the potential of well-managed urbanization in reducing nitrogen pollution in China. The authors conducted a comprehensive analysis of the impacts of urbanization on nitrogen pollution from various sources and forms. They estimated the potential reductions in nitrogen emissions to air and

water bodies and evaluated the feasibility and cost-benefit of these reductions. There are some concerns that should be addressed to enhance the overall quality of the paper.

1. The authors discussed changes in dietary structure in the limitations section but did not mention the limitations of the methods employed in this study. It is better to conduct sensitivity analyses or ablation studies to assess the uncertainties and limitations associated with your methodology. Integrating such analyses would enhance the credibility of your findings.
2. The authors considered the feasibility of measures in terms of policy impact but did not discuss the potential challenges in implementing well-managed urbanization to reduce nitrogen pollution. It would be valuable to explore whether there are potential socioeconomic or institutional barriers that could hinder the widespread adoption of these measures and to provide insights into the challenges and considerations related to implementation.
3. The authors proposed reducing nitrogen pollution through population migration to cities, large-scale cropland reclamation, and the coupling of agriculture and animal husbandry. However, it is also important to consider and discuss the potential adverse impacts of these measures in the paper.
4. In lines 113-115, it is mentioned, "Although 1.3 Mha of cropland area would be occupied due to urban expansion, a total area of 6.9 Mha could be reclaimed from rural homesteads and converted to cropland." I would like to know if the implementation costs of this measure have been considered, such as those related to relocation, compensation, and the cost of reconstructing houses.
5. In lines 169-172, it states, "Agricultural systems dominate Nr losses at the national scale, and large-scale farming, manure recycling fully and emission abatement measures using substantially reduce losses from agricultural systems." When designing the scenario, the authors should provide additional information to clarify whether the "large-scale farming" and "relocating livestock" mentioned in the paper involve changes in the structure of both cropland and livestock production (such as different types of crops and animals).
6. In lines 250-252, it states, "Even if cropland and livestock ... would be reduced due to the reduction of nitrogen input and manure recycling." It appears that it should be "increase in manure recycling" instead of "reduction" to accurately convey the intended meaning.
7. In line 475, the words "urban expansion and land reclamation" are presented in isolation and do not form a distinct subsection or a complete sentence.
8. In lines 516-518, it is stated that "DN represents the crop demand for Nr, defined as the sum of actual synthetic fertilizer inputs and manure Nr returned to cropland in 2017." However, in equation (4), there is a superscript of 2050, which is inconsistent with the statement. Please clarify it.
9. In lines 507-518, you represent crop nitrogen demand (DN) as the sum of actual synthetic fertilizer inputs and manure inputs and then use DN to calculate livestock relocation potential. However, it is well known that synthetic fertilizer input in China is often excessive. In this calculation, it would be more reasonable to use optimal fertilizer application rates instead of actual inputs.
10. In equation (5), the term "20*20" is used, but it is not explained in the text. Please clarify its meaning.
11. In lines 541-544, it is mentioned that "we assume that crop-livestock coupled would reduce 50% NH₃ in manure management..." In lines 570-575, it is stated, "We cited the research findings ... can achieve a 60% reduction in NH₃ emissions from the livestock farming process in China." There appears to be an inconsistency in the percentage reduction mentioned (50% vs. 60%). Please provide clarification and ensure consistency in your explanation.
12. In lines 577-578, it is mentioned that "Since there is no direct report on the relationship between farm size and other Nr loss to water." In fact, there are published articles that focus on the relationship between farm size and nitrate loss in rivers. Please refer to *Guo, Hao, et al. "Farm size increase alters the contribution of land use types to sources of river sediment." *Agriculture, Ecosystems & Environment* 354 (2023): 108566, and consult additional literature. If the authors think that the relevant data is insufficient to be cited in the paper, they should at least discuss it in the discussion section.
13. The repeated citation format "48494849" in line 590 needs to be corrected.

Reviewer #3 (Remarks to the Author):

This paper explored the relationships between urbanization and nitrogen pollution in China, and concluded that well-managed urbanization could halve nitrogen pollution. This paper is of interest to environmental scientists, ecologists, and urban planners. Nonetheless, I think that this work requires some revisions since the article lacks some points to be discussed in this context.

1)What is well-managed urbanization? The abstract and method did not propose the definition of 'well-managed urbanization', which is one of the key words in this paper.

2)The influencing factors of regional differences in urbanization in China are different from those at the national level. Different regions have different urbanization rates and characteristics. Is it reasonable to apply the same prediction model for different regions with distinct urbanization characteristics?

3)The explanation of the predicted results is too lengthy, which needs further refining.

4)The regional differences in urbanization in China are influenced by various factors such as natural and geographical conditions, national and local policies, and regional economic growth differences. The analysis and discussion are too general to reflect the quantitative relationship between urbanization and nitrogen pollution under the comprehensive influence of these factors.

5)Fig.1 and 2 lack scale bars and north arrows.

Response to Referees

We would like to express our gratitude to the reviewers for providing valuable feedback that has enabled us to strengthen the scientific rigor of our study. We have conducted comprehensive revisions of our manuscript. These amendments include the integration of an uncertainty analysis concerning our model simulation techniques, an expanded discussion regarding the impact of PM_{2.5}, an enhanced examination of policy recommendations, and the assimilation of zoning strategies pertinent to policy ramifications, among others. Below, please find detailed accounts of our responses and the corresponding modifications made in relation to each comment.

Referee #1

Comment	Response and Change
Deng and colleagues present a comprehensive method coupling N loss prediction and cost-benefit analysis in order to assess the role of well-managed urbanization in N pollution mitigation. The work is solid and the manuscript is well organized. Nevertheless, there are a few issues which need to be addressed, I elaborate these aspects upon which my assessment is based.	Thank you for the comments. The paper has been revised in line with your recommendations. We hope the revised version alleviates any concerns you may have had.
(1) There are two threads presented in this study. The first part is the changes prediction and N loss reduction with well-managed urbanization. The second part is the cost-benefit analysis of N pollution abatement. They are strangely connected by air PM 2.5. The reduction of N loss could cause lots of impacts, such as the decrease of eutrophication, ozone damage, photo-chemical smog, health damage and other damage. Why only the PM 2.5 is highlighted?	We appreciate your thoughtful observation regarding the two threads presented in our study. We concur that curtailing nitrogen losses can have far-reaching consequences, including but not limited to air and water pollution, biodiversity diminution, and climate change. Within this spectrum of impacts, air pollution—predominantly attributed to PM_{2.5}, with a marginal contribution from ozone (Gu et al., 2021)—poses significant threats to human health. Concurrently, water pollution and biodiversity depletion adversely affect ecosystem vitality. In light of these considerations, our benefit assessment encompasses human health, ecosystems, and climate impact. Given that climate change is chiefly reflected in N₂O emissions, our analysis delves deeper into PM_{2.5} pollution and nitrogen losses affecting aquatic environments. PM_{2.5} stands as a paramount environmental precursor to premature mortality globally (IHE, 2020; Gu, 2021). Our focus on the influence of adeptly administered urbanization on reducing PM_{2.5} levels is predicated on the profound ramifications of PM_{2.5}-induced air pollution

	on human health, corroborating its global prominence as a mortality risk factor. To accentuate the pivotal nature of PM_{2.5} within the purview of this study, we have enriched our manuscript with the following elaborations: Gu, B. et al. Abating ammonia is more cost-effective than nitrogen oxides for mitigating PM_{2.5} air pollution. Science 374, 758-762 (2021). Institute, Health Effects. State of Global Air 2020: A Special Report on Global Exposure to Air Pollution and Its Health Impacts. (the USA, Boston, 2020). In Line 266-280: “Curtailing nitrogen losses can have far-reaching consequences, including but not limited to air and water pollution, biodiversity diminution, and climate change. Within this spectrum of impacts, air pollution—predominantly attributed to PM_{2.5}, with a marginal contribution from ozone³⁷—poses significant threats to human health. Concurrently, water pollution and biodiversity depletion adversely affect ecosystem vitality. In light of these considerations, our following benefit assessment encompasses human health, ecosystems, and climate impact. Given that climate change is chiefly reflected in N₂O emissions, our analysis delves deeper into PM_{2.5} pollution and nitrogen losses affecting aquatic environments. PM_{2.5} stands as a paramount environmental precursor to premature mortality globally³⁸. Our focus on the influence of adeptly administered urbanization on reducing PM_{2.5} levels is predicated on the profound ramifications of PM_{2.5}-induced air pollution on human health, corroborating its global prominence as a mortality risk factor. Given the substantial repercussions of PM_{2.5} on human well-being, coupled with its significant presence in urban landscapes, this study emphasizes the shifts in PM_{2.5} concentrations resultant from targeted reductions in Nr emissions.”
(2) Along this line, in cost-benefit analysis, could the impact of Nr loss on human health be fully assessed by only focusing on PM_{2.5} caused various disease? How about the toxicity of NO_x and related damage to lung and central nervous system?	We concur that the health ramifications of Nr emissions extend beyond diseases related to PM_{2.5}, encompassing ailments arising from the toxicity of NO_x compounds. Our choice to use PM_{2.5} concentrations as a quantifiable indicator of health benefits stems from the substantial impact it has on human health and the availability of comprehensive assessment data. PM_{2.5} is universally acknowledged for its severe and far-reaching

	health consequences. The health effects of PM_{2.5} air pollution, which include conditions such as stroke, heart disease, lung cancer, and a range of chronic and acute respiratory diseases, have been meticulously evaluated. These evaluations are employed by the World Health Organization (WHO) for calculating the financial toll of air pollution-related health issues. While we recognize the value in assessing the toxicity of NO_x for a more nuanced understanding of health benefits, we faced modeling and methodological limitations that precluded us from distinctly parsing out the unique toxic effects attributable to NO_x. To address this, we have supplemented our study with an expanded analysis, discussing the potential uncertainties arising from not fully accounting for all the impacts emanating from Nr emissions. In Line 501-513: “We concur that the health ramifications of N_r emissions extend beyond diseases related to PM_{2.5}, encompassing ailments arising from such as the toxicity of NO_x compounds. Our choice to use PM_{2.5} concentrations as a quantifiable indicator of health benefits stems from the substantial impact it has on human health and the availability of comprehensive assessment data. PM_{2.5} is universally acknowledged for its severe and far-reaching health consequences. The health effects of PM_{2.5} air pollution, which include conditions such as stroke, heart disease, lung cancer, and a range of chronic and acute respiratory diseases, have been meticulously evaluated. These evaluations are employed by the World Health Organization (WHO) for calculating the financial toll of air pollution-related health issues. While we recognize the value in assessing the toxicity of NO_x for a more nuanced understanding of health benefits, we faced modeling and methodological limitations that precluded us from distinctly parsing out the unique toxic effects attributable to NO_x and other potential constrains.”
(3)The authors elaborate the changes with well-managed urbanization, including population migration, agricultural optimization and industrial upgrading. It seems to be a comprehensive optimization of all population, land and industry resources all over the country. Therefore, I wonder the	Thank you for your comment, and we apologize for any lack of clarity in the initial version of our manuscript. We maintain that well-managed urbanization possesses the transformative potential to sculpt the trajectory of social development. Nonetheless, it's crucial to acknowledge that this manuscript doesn't capture every facet of this expansive perspective. Moreover, the dynamics of population migration, agricultural optimization, and industrial upgrading elaborated upon in this document are directly spurred by well-

definition or the boundary of well-managed urbanization. Whether all the changes you mentioned are driven by urbanization? Whether the urbanization can outline all the perspectives for the future social development?	managed urbanization initiatives. To enhance clarity and precision on these points, we have introduced a detailed definition and delineation of "well-managed urbanization" at the outset of the results section. This addition is intended to firmly establish the scope and implications of well-managed urbanization within the context of our study. In Line 97-110: "Urbanization signifies a pivotal demographic transition from rural to urban living, a change that is profoundly shaping contemporary China. When effectively managed, urbanization can serve as a dynamic catalyst for various dimensions of sustainable development by conscientiously balancing environmental stewardship with socioeconomic considerations. This study, however, narrows its focus specifically to the aspects of the urbanization process that are intricately tied to nitrogen management through biophysical changes. This encompasses land intensification, the refinement of agricultural practices, and the expansion of treatment capabilities for domestic and industrial pollution. Therefore, our concept of well-managed urbanization pertains to the governance of land, resources, and the environment throughout the urbanization trajectory, deliberately omitting the management of social and economic aspects. It's worth noting, for example, that urbanization, when leveraged appropriately, has the potential to alleviate poverty and reduce inequality by expanding job prospects, improving living standards, enhancing educational opportunities, and bolstering healthcare services—factors that are outside the scope of our current investigation."
(4) In the results part, the North China Plain is so conspicuous due to high N loss in 2017 and high reduction potential. However, in the Northeast area, where agricultural production holds large proportion in the total production nationwide, and also will experience intensive population reduction, urbanization increasing and cropland change according your prediction shown in Fig. 1, has still low TN loss potential (Fig. 2c). Why? More analysis and discussion for spatial heterogenicity of N loss is	Thank you for your insightful observations. It's true that while both the North China Plain and the Northeast Plain are pivotal agricultural hubs, they diverge significantly in their production methodologies and efficiencies in fertilizer usage. The North China Plain is particularly prone to excessive N fertilizer application, a consequence of its intensive double cropping system and fragmented farm sizes. Conversely, the Northeast, with its single cropping practice and expansive farm sizes, tends to mitigate N fertilizer overuse, thereby enhancing nitrogen use efficiency. Consequently, impending urbanization is projected to have a more subdued role in curtailing N losses in the Northeast compared to its influence in the North China Plain.

needed combining with actual industries, agriculture (such as coupled crop-livestock production), economy, population and policy, which could also provide implications for practice.	In our commitment to thoroughly investigate this disparity, we've instituted a "Zoning strategy" within our study. The ensuing section provides a detailed examination of the variances between the Northeast and North China, articulating the unique agricultural practices and resultant environmental implications in these regions: In Line 435-461: “... Agricultural practices, urban growth, and urbanization management strategies in China exhibit considerable regional diversity. Regions such as the North China Plain, the Middle and Lower Yangtze River Plain, and the Sichuan Basin are anticipated hotspots of swift urbanization and are crucial food-producing areas. As of 2017, these areas primarily relied on small-scale farming characterized by excessive nitrogen fertilizer usage and suboptimal NUE levels of 20%-40%⁴⁴. Additionally, these regions are known for monogastric animal farming, leading to lower NUEs compared to northern areas where ruminant farming prevails⁴⁵. These pecific regions are instrumental in curtailing nitrogen pollution, including NH₃ and N₂O emissions as well as runoff losses. They present the most favorable benefit-to-cost ratio for intervention measures, underlining the urgency for expeditious land system reforms, upscaling to large-scale production, and embracing integrated crop-livestock methodologies. ..... Conversely, the Northwestern and Northeastern regions, despite their sparser populations, have triumphed in establishing extensive agricultural production, achieving NUEs above 50%—a feat unmatched by other regions in China⁴⁴. With population dips and diminished urban stressors on the horizon, these territories require sustained land reforms and agricultural fine-tuning, although their prospects for nitrogen pollution mitigation are relatively constrained. ”
(5) I noticed that authors opted for including so-called “S2 Comparison of different water quality model” in supporting information. Including discussion item in this section seems not appropriate and therefore I invite the authors to critically evaluate whether this	Thanks for your comments. We have moved the “S2 Comparison of different water quality model” to the main document, now included in the Limitation section, with the aim of addressing uncertainties related to the supplementary method.

information could be shifted to the main document.	
Generally speaking the manuscript is well organized and formatting is mostly adequate. Nevertheless, I spotted several mistakes listed as follows. Therefore, I recommend the authors to re-check the whole manuscript. Line 590, the reference is cited repeatedly. Equation 9, line 588, I believe it should include a NO₃ discharge (ND) rather than all NE.	Apologies for the errors in the manuscript. We've thoroughly reviewed the entire document, including citations and formulas. Please review the tracked manuscript. We have added equation 10 as follow : $NE_g^{2050} = \exp(\alpha \times \ln NE_g^{2017} - \alpha \times \ln NE_g^{2050}) \times NE_g^{2017} \quad (9)$ $ND_g^{2050} = \exp(\beta \times \ln ND_g^{2017} - \beta \times \ln ND_g^{2050}) \times ND_g^{2017} \quad (10)$ NE_g and NE_g represent industrial NO_x emission and NO₃⁻ discharge in the gth province (kg) respectively, α and β refers to estimated coefficients from Xu et al^{55, 56}.

Referee #2

Comment	Response and Change
The manuscript titled "Well-managed urbanization could halve nitrogen pollution in China" aims to investigate the potential of well-managed urbanization in reducing nitrogen pollution in China. The authors conducted a comprehensive analysis of the impacts of urbanization on nitrogen pollution from various sources and forms. They estimated the potential reductions in nitrogen emissions to air and water bodies and evaluated the feasibility and cost-benefit of these reductions. There are some concerns that should be addressed to enhance the overall quality of the paper.	Thank you for the comments. The paper has been revised in line with your recommendations. We hope the revised version alleviates any concerns you may have had.
1. The authors discussed changes in dietary structure in the limitations section but did not mention the limitations of the methods employed in this study. It is	Thank you for your insightful comments. Urbanization significantly impacts dietary structures through various factors such as education, culture, employment, among others. For instance, as urbanization levels rise, there's typically an increase in the

better to conduct sensitivity analyses or ablation studies to assess the uncertainties and limitations associated with your methodology. Integrating such analyses would enhance the credibility of your findings.

average educational years, which often leads to healthier dietary choices. Conversely, heightened urbanization tends to reduce the populace engaged in manual labor, leading to a reduced caloric demand. However, gauging the exact effects on outcomes via sensitivity analyses or ablation studies is indeed challenging. It's crucial to note that this study predominantly emphasizes well-managed urbanization, which is initially defined in terms of biophysical changes, sidelining social and economic aspects. As such, a comprehensive exploration of the implications of urbanization on future agricultural production structures and nitrogen pollution via dietary structure remains outside the purview of this study, but will be a subject of our future research endeavors. ***Acknowledging your feedback, we have now incorporated a section addressing the uncertainties tied to multi-model coupling under the limitations. We believe this will offer readers a more holistic grasp of the methodological constraints presents in our research.***

In Line 515-535: “This study utilized a sophisticated multi-modeling methodology to evaluate both air and water pollution. In particular, the PM_{2.5} concentrations appraised for the year 2017 demonstrated a strong alignment with the actual measured values, as evidenced by a correlation coefficient ranging from 0.54 to 0.73 (Fig. S1). However, the validation of national-scale water quality models remains a complex endeavor, largely due to the limited availability of comprehensive time series data and information across various river sections. Consequently, we juxtaposed Nr water pollution levels with findings from preceding studies. Our analysis inferred that approximately 15.6 Tg of Nr was deposited into aquatic systems, a figure calculated through the CHANS model. This estimate closely mirrors the findings presented by the IMAGE-GNM (15.4 Tg)⁴⁸ and DNDC models (14.5 Tg)¹⁰. Moreover, our projection of Nr discharge into the ocean, derived using the WNF model, was approximately 6.8 Tg. This figure is conservative when compared to the 11.7 Tg suggested by the IMAGE-GNM model but exceeds the estimates put forth by the MARINA model (0.83-1.18 Tg)⁴⁹ and the official report from China, which stands at 2.3 Tg⁵⁰. Despite the inherent uncertainties surrounding the extent of Nr contributions to marine environments, we ascertain our estimation outcomes to be within reasonable bounds. It's pivotal to note that this research primarily aims to discern the effects of proficiently orchestrated

	urbanization on nitrogen flows into rivers and oceans, rather than to quantify nitrogen loss on a national scale. The synergistic application of CHANS and WNF models ensures a harmonious simulation of nitrogen transit processes, implying that variations in nitrogen inputs into watercourses and seas are solely attributable to shifts in emission origins consequent to well-managed urbanization.”
2. The authors considered the feasibility of measures in terms of policy impact but did not discuss the potential challenges in implementing well-managed urbanization to reduce nitrogen pollution. It would be valuable to explore whether there are potential socioeconomic or institutional barriers that could hinder the widespread adoption of these measures and to provide insights into the challenges and considerations related to implementation.	Thank you for raising this point. In the revised manuscript, we have introduced a section titled "Challenges and Policy Implications." This section aims to highlight potential socioeconomic or institutional challenges and analyze the opportunities and feasibility of these measure and how improve policy to address these challenges. In Line 371-433: “Challenges and policy implications This study introduces a series of measures to achieve effective resource and environmental management during urbanization process, alongside the goal of halving nitrogen pollution. However, the implementation of these measures presents institutional and socioeconomic challenges. Firstly, the fragmentation of rural land ownership in China inhibits the development of large-scale farming, while land consolidation projects and livestock relocation require significant financial investments, with relatively low agricultural production benefits and weak incentives for producers. Additionally, despite the potential for high economic returns in urban industries to drive industrial upgrading and pollution control, this also necessitates standardized and regulated systems. Below, we will analyze the opportunities and feasibility of these measure and how improve policy to address these challenges. Rural land system reform. The Chinese government has promoted the scale of cropland tenure and the reclamation of homesteads through rural land system reform. With urbanization and rural aging, an increasing number of farmers voluntarily leasing their cropland to new farming models, which has reached 37 Mha as of 2019⁴⁰. Simultaneously, the hollowing of villages has become widespread in central and western China as the younger generation has moved to urban areas. In this context, over the past decade, the potential rural reclamation area has exceeded 4000 ha⁴¹.

However, additional efforts will be needed to address potential challenges such as land quality monitoring and maintenance of supporting infrastructure in response to potential land quality degradation resulting from the separation of land ownership and management rights.

Cropland consolidation investment. An annual investment of approximately US\$21 billion has been dedicated to transforming fragmented croplands into large-scale, high-standard farms. The Chinese government has already initiated a cropland consolidation initiative with the ambitious goal of achieving 72 Mha of high-standard croplands equipped with modern management and facilities in the forthcoming decades⁴². These efforts align with our urbanization projections and underscore the importance of conducting suitability assessments for high-standard farmland construction while avoiding high-cost, non-scalable areas. To enhance this endeavor, the government should refine the financing model by utilizing land tenure as collateral in land reform and increasing subsidized interest rates for high-standard farmland construction. Collaboration with institutions like the Agricultural Development Bank can further stimulate investments in cropland.

Subsidy for livestock relocation. For livestock relocation, the Chinese central government requires newly-built livestock farms to be surrounded by a certain area of croplands for manure recycling. This ensures livestock relocation is not required. Where livestock farms have already been built, livestock relocation costs could be subsidized by government. It should be noted that cross-province relocation of livestock only accounts for 8% of livestock needed to be relocated (Extended data Fig. 4), suggesting the bulk of the relocation is estimated to occur within provinces, increasing the feasibility of livestock relocation due to farmers may not want to move far away from home. With the increase of large-scale croplands, having both crop and livestock production located close to each other becomes more viable for rural households, further aided by a reduction of implementation barriers by governments. In any case, subsidies supporting the relocation of livestock are vital, and central government has recently provided subsidies worth about US\$ 750 million per county to improve the coupling of livestock and crop production as a measure to reduce

	environmental pollution originating from manure management in 100 demonstration counties⁴³. Urban policy integration. Upgrading industrial and waste treatment facilities are essential measures responding to an increase in urban population. A continued focus on the implementation of the Clean Air Act and the Clean Water Act would further aid the reduction of Nr losses from the human system. Meanwhile, the newly implemented policy “Co-reduction of pollution and carbon emission” would link efforts for pollution control with the target of carbon neutrality (NetZero) while pursuing urbanization objectives. This will place a focus on measures which can achieve both goals in a cost-effective and overall cost-beneficial manner. These policy goals have already put Nr reductions on a good pathway, enhancing the feasibility of halving Nr pollution in the context of urbanization. Further actions to include measures and policies with relevance for Nr pollution reduction into wider considerations, including linking to climate change mitigation and NetZero targets, would contribute to achieving a range of SDGs in China and beyond. ..... ”
3. The authors proposed reducing nitrogen pollution through population migration to cities, large-scale cropland reclamation, and the coupling of agriculture and animal husbandry. However, it is also important to consider and discuss the potential adverse impacts of these measures in the paper.	Thanks for your comments. We recognize that uncontrolled urbanization can lead to various problems, as highlighted in the introduction, including issues like inadequate water treatment infrastructure resulting in significant pollution and health consequences. Nevertheless, when urbanization is effectively managed, it can yield national-scale benefits, such as an overall reduction in nitrogen pollution. It's worth noting that the only potential drawback is the risk of increased pollution in megacities, as we have included in the revised manuscript. In Line 448-454: “Eastern China, in the throes of accelerated urbanization and industrial expansion, is critical in the battle against nitrogen pollution, especially concerning NO_x emissions. The burgeoning populations in coastal urban conglomerates, notably the Beijing-Tianjin-Hebei cluster, predict a persistent surge in pollution emissions. In spite of the overarching developmental gains, this area is bracing for the urbanization-induced pollution repercussions. It is imperative for the

	region to steadfastly enhance domestic pollution treatment capabilities and enforce stringent emission controls, potentially through industrial evolution and structural overhauls.”
4. In lines 113-115, it is mentioned, "Although 1.3 Mha of cropland area would be occupied due to urban expansion, a total area of 6.9 Mha could be reclaimed from rural homesteads and converted to cropland." I would like to know if the implementation costs of this measure have been considered, such as those related to relocation, compensation, and the cost of reconstructing houses.	Thank you for raising this concern, it is something we have considered when estimating our costs. Firstly, in this study, we focus on homesteads left behind by farmers who have relocated to cities, not those who relocate through centralized living within villages, which is why we did not estimate relocation and reconstructing costs. Secondly, we did not factor in the subsidies that farmers forgo when giving up their homesteads, as these are often used to purchase new dwellings in cities. We assume that their migration is a result of natural process of urbanization, not a well-managed measure. To make the above-mentioned points clearer, we added following sentence in Line 753-756. “We did not factor in the cost of homeownership for farmers relocating to urban areas, as we assume their migration is part of natural process of urbanization, not a well-managed measure, and their implementation costs have already been balanced through market regulations.”
5. In lines 169-172, it states, "Agricultural systems dominate Nr losses at the national scale, and large-scale farming, manure recycling fully and emission abatement measures using substantially reduce losses from agricultural systems." When designing the scenario, the authors should provide additional information to clarify whether the "large-scale farming" and "relocating livestock" mentioned in the paper involve changes in the structure of both cropland and livestock production (such as different types of crops and animals).	Thanks for your valuable comments. We have added related information in the Method. In Line 578: “The planting structure of crops in this process remains consistent with year 2017.” In Line 583-585: “The relocation process only takes into account livestock from agricultural areas, and the livestock production structure remains consistent with the year 2017.”
6. In lines 250-252, it states, "Even if cropland and livestock ... would be reduced due to the reduction of	We apologize for this error. We have made a revision to this sentence in Line 227-229:

nitrogen input and manure recycling." It appears that it should be "increase in manure recycling" instead of "reduction" to accurately convey the intended meaning.	“Even if cropland and livestock N₂O emission factors remain constant, N₂O emissions would be reduced due to reduced nitrogen input and increased manure recycling.”
7. In line 475, the words "urban expansion and land reclamation" are presented in isolation and do not form a distinct subsection or a complete sentence.	We apologize for this error. We have deleted these words.
8. In lines 516-518, it is stated that "DN represents the crop demand for Nr, defined as the sum of actual synthetic fertilizer inputs and manure Nr returned to cropland in 2017." However, in equation (4), there is a superscript of 2050, which is inconsistent with the statement. Please clarify it.	Thanks for your comments. In line with your subsequent comment, we have revised this equation and its explanation as following: In Line 593-600: $DN_i = F_i^{2050} + M_i \quad (4)$ where LCP_{max} represents the maximum manure Nr carrying capacity of cropland (around 144 kg N ha⁻¹)⁵¹, LI_i represents the total manure Nr (kg) in the county where the ith grid is located (kg), LCA_i represents the areas of large-scale cropland where the ith grid is located (km²), DN_i represents the crop demand for Nr, calculated as the sum of optimal fertilizer application rates (F_i, kg ha⁻¹) and manure Nr returned to cropland (M_i, kg ha⁻¹) in 2050. We assume that F_i equals the synthetic fertilizer inputs after cropland scale-up, as estimated in equation (7).
9. In lines 507-518, you represent crop nitrogen demand (DN) as the sum of actual synthetic fertilizer inputs and manure inputs and then use DN to calculate livestock relocation potential. However, it is well known that synthetic fertilizer input in China is often excessive. In this calculation, it would be more reasonable to use optimal fertilizer application rates	Thanks for your valuable comment. We agree that it's more appropriate to use the optimal fertilizer application rates in this context rather than the actual amount of fertilizer applied. We assume the optimal fertilizer application rates equals the synthetic fertilizer inputs after farm size increase. This assumption is rooted in the fact that the widespread adoption of new technologies, like 4R nutrient management and soil testing on large-scale farms, has led to a substantial reduction in fertilizer usage, bringing it in line with the crop's actual demand. See answer to previous question for revision details

instead of actual inputs.	
10. In equation (5), the term "20*20" is used, but it is not explained in the text. Please clarify its meaning.	Thanks for your comments. We have revised the explanation of the equation. In Line 609-610: "where N_i is the number of overloads grids within surrounding 400 (20×20) grids and ith grids in the central of 400 grids. "
11. In lines 541-544, it is mentioned that "we assume that crop-livestock coupled would reduce 50% NH3 in manure management..." In lines 570-575, it is stated, "We cited the research findings ... can achieve a 60% reduction in NH3 emissions from the livestock farming process in China." There appears to be an inconsistency in the percentage reduction mentioned (50% vs. 60%). Please provide clarification and ensure consistency in your explanation.	We apologize for this error. We have revised 50% to 60%.
11. In lines 577-578, it is mentioned that "Since there is no direct report on the relationship between farm size and other Nr loss to water." In fact, there are published articles that focus on the relationship between farm size and nitrate loss in rivers. Please refer to *Guo, Hao, et al. "Farm size increase alters the contribution of land use types to sources of river sediment." Agriculture, Ecosystems & Environment 354 (2023): 108566, and consult additional literature. If the authors think that the relevant data is insufficient to be cited in the paper, they should at least discuss it in the discussion section.	We appreciate your insightful observation regarding the connection between farm size and Nr loss to water. We concur that expanding farm sizes in hilly areas may lead to increased soil erosion and Nr loss to water. However, there is a scarcity of quantitative studies on these aspects when other variables are held constant. Consequently, based on your comment, we have added a relevant discussion to further complement this topic in the paper. In Line 464-472: "..... This terrain, representing 10% of the country's arable land, faces challenges in transitioning to large-scale farming by 2050 due to its topographical constraints (Extended Data Fig. 3b). Compounding these challenges are the poorer soil conditions relative to other regions. It's crucial to recognize that expanding agricultural landholdings here may inadvertently intensify soil erosion and nitrogen depletion⁴⁶. To bolster nitrogen utilization efficiency and fortify food security, innovative strategies—such as altering crop compositions or strategic relocation of cultivable land—are worth investigating. Consequently, under existing policies, the scope for curtailing nitrogen emissions in these Southwestern highlands is

	circumscribed.”
13. The repeated citation format "48494849" in line 590 needs to be corrected.	Apologies for the errors in the manuscript. We've corrected the error you mentioned and thoroughly reviewed the entire document.

Referee #3

Comment	Response and Change
This paper explored the relationships between urbanization and nitrogen pollution in China, and concluded that well-managed urbanization could halve nitrogen pollution. This paper is of interest to environmental scientists, ecologists, and urban planners. Nonetheless, I think that this work requires some revisions since the article lacks some points to be discussed in this context.	Thank you for the comments. The paper has been revised in line with your recommendations. We hope the revised version alleviates any concerns you may have had.
1)What is well-managed urbanization? The abstract and method did not propose the definition of ‘well-managed urbanization’, which is one of the key words in this paper.	Thank you for this valuable comment. We have added a section to define well-managed urbanization at the beginning of result section. In Line 97-110: “Urbanization signifies a pivotal demographic transition from rural to urban living, a change that is profoundly shaping contemporary China. When effectively managed, urbanization can serve as a dynamic catalyst for various dimensions of sustainable development by conscientiously balancing environmental stewardship with socioeconomic considerations. This study, however, narrows its focus specifically to the aspects of the urbanization process that are intricately tied to nitrogen management. This encompasses land intensification, the refinement of agricultural practices, and the expansion of treatment capabilities for domestic and industrial pollution. Therefore, our concept of well-managed urbanization pertains to the governance of land, resources, and the environment throughout the urbanization trajectory, deliberately omitting the direct management of social and economic aspects. It's worth noting, for example, that urbanization, when leveraged

	appropriately, has the potential to alleviate poverty and reduce inequality by expanding job prospects, improving living standards, enhancing educational opportunities, and bolstering healthcare services—factors that are outside the scope of our current investigation.”
2)The influencing factors of regional differences in urbanization in China are different from those at the national level. Different regions have different urbanization rates and characteristics. Is it reasonable to apply the same prediction model for different regions with distinct urbanization characteristics?	Thanks for your comments. The simulation of urbanization process is carried out on a national scale, with model parameters, such as urbanization rate and farmland size, varying from over 2800 counties. In order to examine population changes, agricultural models, and urbanization management strategies across different regions, we have introduced a section titled "Zoning Strategy" in the revised manuscript. We hope that this addition will better illustrate how well-managed urbanization can contribute to the environmental governance of resources in different regions. In Line 435-461: “Zoning strategy. Agricultural practices, urban growth, and urbanization management strategies in China exhibit considerable regional diversity. Regions such as the North China Plain, the Middle and Lower Yangtze River Plain, and the Sichuan Basin are anticipated hotspots of swift urbanization and are crucial food-producing areas. As of 2017, these areas primarily relied on small-scale farming characterized by excessive nitrogen fertilizer usage and suboptimal NUE levels of 20%-40%⁴⁴. Additionally, these regions are known for monogastric animal farming, leading to lower NUEs compared to northern areas where ruminant farming prevails⁴⁵. These pecific regions are instrumental in curtailing nitrogen pollution, including NH₃ and N₂O emissions as well as runoff losses. They present the most favorable benefit-to-cost ratio for intervention measures, underlining the urgency for expeditious land system reforms, upscaling to large-scale production, and embracing integrated crop-livestock methodologies. Eastern China, in the throes of accelerated urbanization and industrial expansion, is critical in the battle against nitrogen pollution, especially concerning NO_x emissions. The burgeoning populations in coastal urban conglomerates, notably the Beijing-Tianjin-Hebei cluster, predict a persistent surge in pollution emissions. In spite of the overarching developmental gains, this area is bracing for the urbanization-induced

	pollution repercussions. It is imperative for the region to steadfastly enhance domestic pollution treatment capabilities and enforce stringent emission controls, potentially through industrial evolution and structural overhauls. Conversely, the Northwestern and Northeastern regions, despite their sparser populations, have triumphed in establishing extensive agricultural production, achieving NUEs above 50%—a feat unmatched by other regions in China⁴⁴. With population dips and diminished urban stressors on the horizon, these territories require sustained land reforms and agricultural fine-tuning, although their prospects for nitrogen pollution mitigation are relatively constrained. ”
3)The explanation of the predicted results is too lengthy, which needs further refining.	Thanks for your comments. We have refined the Results, especially the “Reduction of total Nr loss”. Kindly review the new manuscript.
4)The regional differences in urbanization in China are influenced by various factors such as natural and geographical conditions, national and local policies, and regional economic growth differences. The analysis and discussion are too general to reflect the quantitative relationship between urbanization and nitrogen pollution under the comprehensive influence of these factors.	Thanks for your valuable comments. We have added a section titled “Zoning strategy” to analyze population change, agriculture model, and urbanization management strategies in various regions. Please check response to question 2 for details.
5)Fig.1 and 2 lack scale bars and north arrows.	Thanks for your comments. We have added scale and north arrows in the Fig.1 and 2.

Fig. 1 | Population, land use and agricultural change with urbanization.

These figures demonstrate changes in population, land use and agricultural system with urbanization from 2017 to 2050. **a**, Geo-distribution of population change. **b**, Change of population. **c**, Geo-distribution of land use change, with the same colors as in **d**. **d**,

Quantity change of land use. **e**, Geo-distribution of urbanization change. **f**, Geo-distribution of cropland change. **g**, Geo-distribution of livestock change. The base map is applied without endorsement from GADM data (<https://gadm.org/>).

Fig. 2 | Spatial variation of nitrogen loss with urbanization.

a, Distribution of total reactive nitrogen (TN) losses in 2017. **b**, Distribution of predicted TN losses in 2050 assuming an urbanization level of 80%. **c**, Distribution of the change in TN losses during the 2017-2050 period. **d**, National NH_3 , NO_x , N_2O and NO_3^- loss in 2017 and 2050. TN losses include NH_3 , NO_x , N_2O and NO_3^- loss. We use NO_3^- to

	represent the N_r loss via runoff and leaching, given that most of the other dissolve forms of N_r would be converted to NO_3^- during moving with water. The base map is applied without endorsement from GADM data (https://gadm.org/).
--	---

REVIEWERS' COMMENTS

Reviewer #1 (Remarks to the Author):

The revised manuscript has been streamlined and improved. My major comments have been addressed. However, there are still a number of minor grammatical issues throughout (e.g. figure format), and these should be addressed prior to final publication.

Reviewer #2 (Remarks to the Author):

I have carefully reviewed your submitted manuscript and appreciate your thorough responses and revisions. I believe you have addressed my previous review comments effectively and made valuable amendments to the manuscript. You have added discussions regarding the uncertainties in some of the methods, challenges and opportunities in policy implementation, and detailed optimizations.

Your work explores the feasibility of reducing nitrogen pollution through well-managed urbanization, offering a novel approach to mitigating nitrogen emissions. Your research suggests that by 2050, China could potentially reduce nitrogen pollution by approximately 50%, resulting in significant environmental and economic benefits. I agree to publish in the current form.

Reviewer #3 (Remarks to the Author):

The authors have conducted a comprehensive revision of this manuscript according to my comments.